# Methylphosphonate-driven methane formation and its link to primary production in the oligotrophic North Atlantic

Jan N. von Arx [1] ✉, Abiel T. Kidane[1], Miriam Philippi [1,2], Wiebke Mohr [1], Gaute Lavik[1], Sina Schorn [1], Marcel M. M. Kuypers [1] & Jana Milucka [1]

Methylphosphonate is an organic phosphorus compound used by microorganisms when phosphate, a key nutrient limiting growth in most marine surface waters, becomes unavailable. Microbial methylphosphonate use can result in the formation of methane, a potent greenhouse gas, in oxic waters where methane production is traditionally unexpected. The extent and controlling factors of such aerobic methane formation remain underexplored. Here, we show high potential net rates of methylphosphonate-driven methane formation (median 0.4 nmol methane $L^{-1} d^{-1}$) in the upper water column of the western tropical North Atlantic. The rates are repressed but still quantifiable in the presence of in-situ or added phosphate, suggesting that some methylphosphonate-driven methane formation persists in phosphate-replete waters. The genetic potential for methylphosphonate utilisation is present in and transcribed by key photo- and heterotrophic microbial taxa, such as *Pelagibacterales*, *SAR116*, and *Trichodesmium*. While the large cyanobacterial nitrogen-fixers dominate in the surface layer, phosphonate utilisation by *Alphaproteobacteria* appears to become more important in deeper depths. We estimate that at our study site, a substantial part (median 11%) of the measured surface carbon fixation can be sustained by phosphorus liberated from phosphonate utilisation, highlighting the ecological importance of phosphonates in the carbon cycle of the oligotrophic ocean.

The open ocean (>2000 m water depth) emits up to an estimated 0.5 mmol $m^{-2} yr^{-1}$ of the biogenic greenhouse gas methane ($CH_4$) to the atmosphere[1], which amounts to ca. 5–23% of the total estimated emissions from the marine environment[1]. Intriguingly, the underlying cause of these emissions is to a large extent methane supersaturation in oxic surface waters[2–4], which has been dubbed the so-called 'marine methane paradox'[5]. The paradoxical nature of this phenomenon stems from the fact that biological methane production by methanogenic archaea is a strictly anaerobic process[6]. While some methanogenesis can occur in discrete anoxic microniches such as fish and zooplankton guts and/or particles[7–9], the prevailing mechanisms behind the marine methane paradox are likely truly aerobic methane formation processes. In recent years, diverse aerobic bacteria have been shown to possess the capacity to form methane both from methylated organic compounds (e.g. methylphosphonate[10], methylamine[11], dimethylsulfoniopropionate[12] or methyl radicals[13]) or dissolved inorganic carbon[14–16].

Methylphosphonate utilisation has been identified as the underlying cause of methane supersaturation in the oligotrophic gyres of the North Pacific and Atlantic, where organic phosphorus compounds are used as a source of phosphorus due to the unavailability of inorganic phosphate (Pi)[10,17,18]. Phosphonates are characterised by a stable

[1]Max Planck Institute for Marine Microbiology, Bremen, Germany. [2]Alfred Wegener Institute Helmholtz Centre for Polar and Marine Research, Bremerhaven, Germany. ✉e-mail: jarx@mpi-bremen.de

carbon-phosphorus bond that can be cleaved by the multisubstrate enzyme carbon-phosphorus lyase (C-P lyase)[19]. The *phnJ* gene, encoding for the catalytic subunit[20], is therefore used as a marker gene for phosphonate-utilising microorganisms[21].

Despite the important role of methylphosphonate utilisation in the marine carbon and phosphorus cycles, the process has so far been experimentally confirmed only in a handful of oceanic regions, mainly around station ALOHA in the eastern tropical North Pacific[10,17,22] and in the Sargasso Sea[23]. To date, little is known about the depth distribution of the process in the water column and the physico-chemical factors that control it. The process is thought to be induced under Pi-limitation, which is prevalent in the subtropical gyres but can be found in most of the surface open ocean[24]. One of the most chronically Pi-limited marine regions is the western North Atlantic[18,25,26], a region with low productivity[27,28] that is reliant on recycled nutrients. As such, the usage of organic phosphorus compounds by the microbial community is widespread[23,29,30]. While subsurface methane concentration maxima were observed in the North Atlantic[4,31], their sources are not well constrained, particularly for the tropical region. Additionally, the potential of methylphosphonate for sustaining primary productivity in oligotrophic ocean waters is so far unknown.

During the cruise M161, we investigated aerobic methane formation in the upper 200 metres of the water column east of Barbados using a range of experimental and molecular methods. Our data strongly support the primary role of methylphosphonate as a source of methane in this region with detectable potential net rates across the whole studied water column, including the Pi-rich waters below the deep chlorophyll maximum. We identify the abundant marine bacterial photo- as well as heterotrophic organisms, including *Trichodesmium*, *SAR11*, *SAR116* and *Rhodobacteraceae* as potential candidates behind aerobic methane formation in the western tropical North Atlantic.

## Results and discussion
### Rates of methylphosphonate-driven methane formation decrease with depth
The western North Atlantic is one of the most phosphate (Pi) limited oceanic regions worldwide, with Pi concentrations of 0.2–1 nmol L$^{-1}$ in the surface waters[18,26]. Correspondingly, the surface waters of our study area east of Barbados (Fig. 1a) were depleted in both nitrogen

(NOx below detection) and Pi (below detection; Fig. 2). Dissolved organic phosphorus (DOP) was present throughout the water column at concentrations comparable to e.g. the Sargasso Sea[23,32] (up to 153 nmol L$^{-1}$; Fig. 2d). Satellite measurements of the region revealed elevated chlorophyll *a* (Chl *a*) concentrations in the three southernmost stations (stations 4, 7 and 10; Fig. 1b). This distinction was reflected by a shallower deep chlorophyll maximum (DCM; defined here as the Gaussian area encompassing Chl *a* concentration above 0.35 µg L$^{-1}$) located below the mixed layer in these stations (Fig. 1c; Supplementary Fig. S1). Inorganic N (nitrogen oxides + ammonium) to P (Pi) ratios were not quantifiable for most of the water column above and within the DCM, as concentrations of Pi were consistently below the detection limit (10 nmol L$^{-1}$; Fig. 2e). Below the DCM, inorganic N to Pi ratios increased to excesses of 20, as observed previously in the Sargasso Sea[26], suggesting, based on the Redfield ratio[33], that even the deeper waters remained Pi-limited. This pattern was also observed within the particulate organic fraction in the POC:PON:POP ratios (Supplementary Figs. S2, S3). Carbon fixation rates, measured as the incorporation of dissolved inorganic carbon ($^{13}$C-DIC) into biomass, were comparable among the upper three incubation depths and usually in the range of 200 and 600 nmol C L$^{-1}$ d$^{-1}$ (Fig. 2f). Rates were still measurable below the DCM, but reduced ca. 100-fold, likely due to light limitation and a lower abundance of primary producers with increasing depth[34,35]. Carbon fixation rates were consistently higher in the southern "high chlorophyll" stations (Fig. 2f).

Methylphosphonate-driven methane (CH$_4$) formation (methylphosphonate-addition experiment) was quantifiable in all stations and at all investigated depths (10-metre surface depth, an intermediate depth above the DCM, at the DCM and below the DCM). The variability of the methane formation rate between duplicate incubations (from the same station and depth) was comparable to the variability observed across the twelve stations; therefore, rates from all stations were combined for further analyses (Fig. 3). The onset of methane formation in methylphosphonate incubations was mostly immediate and linear over the course of 24 hours of incubation but often followed by exponential increases after 24 hours (Supplementary Note 1; Supplementary Fig. S4). The rates did not appear to vary substantially in response to the day and night cycle (see Supplementary Note 2 for discussion on abiotic photodegradation of methylphosphonate). The highest rates of methane formation were consistently measured in the

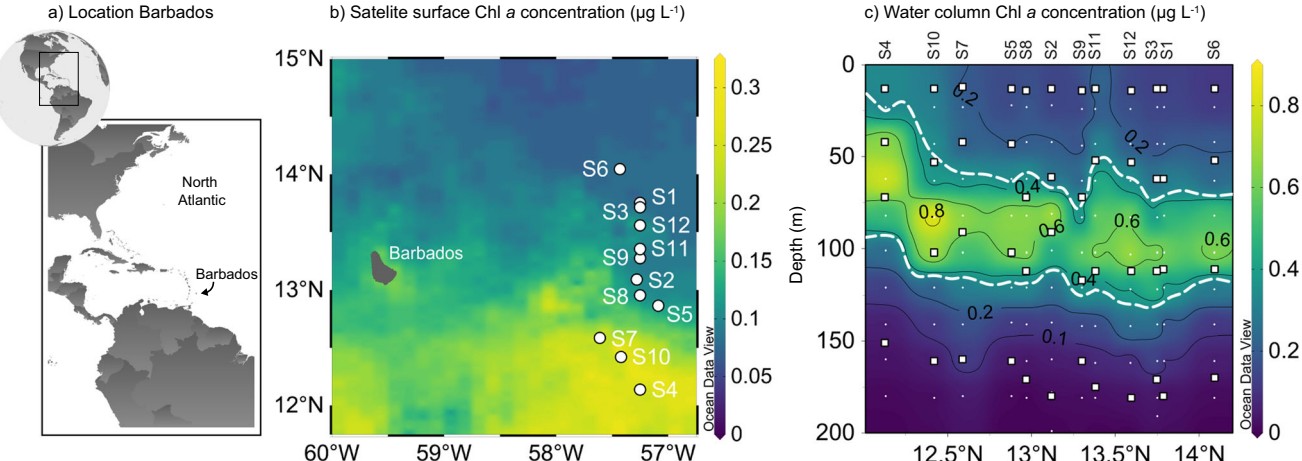

**Fig. 1 | Chlorophyll *a* (Chl *a*) concentrations of the sampling area in the western tropical North Atlantic east of Barbados. a** Sampling site location. **b** Satellite surface Chl *a* concentrations (see methods) averaged for the duration of the sampling campaign revealed a clear north-south divide, with higher Chl *a* concentrations in the southern stations 4, 7 and 10 compared to the northern ones, all stations are represented as white circles. **c** Water column Chl *a* concentrations for the upper 200 m. The dashed white line outlines the deep chlorophyll maximum (area between 0.35 µg L$^{-1}$ Chl *a*). White squares represent the incubation depths and white points the sampling depths from the profiling CTD cast. Maps were generated in QGIS (**a**), as well as Ocean Data View (Schlitzer, Reiner, Ocean Data View, https://odv.awi.de;[87] **b**). Source data are provided as a source data file.

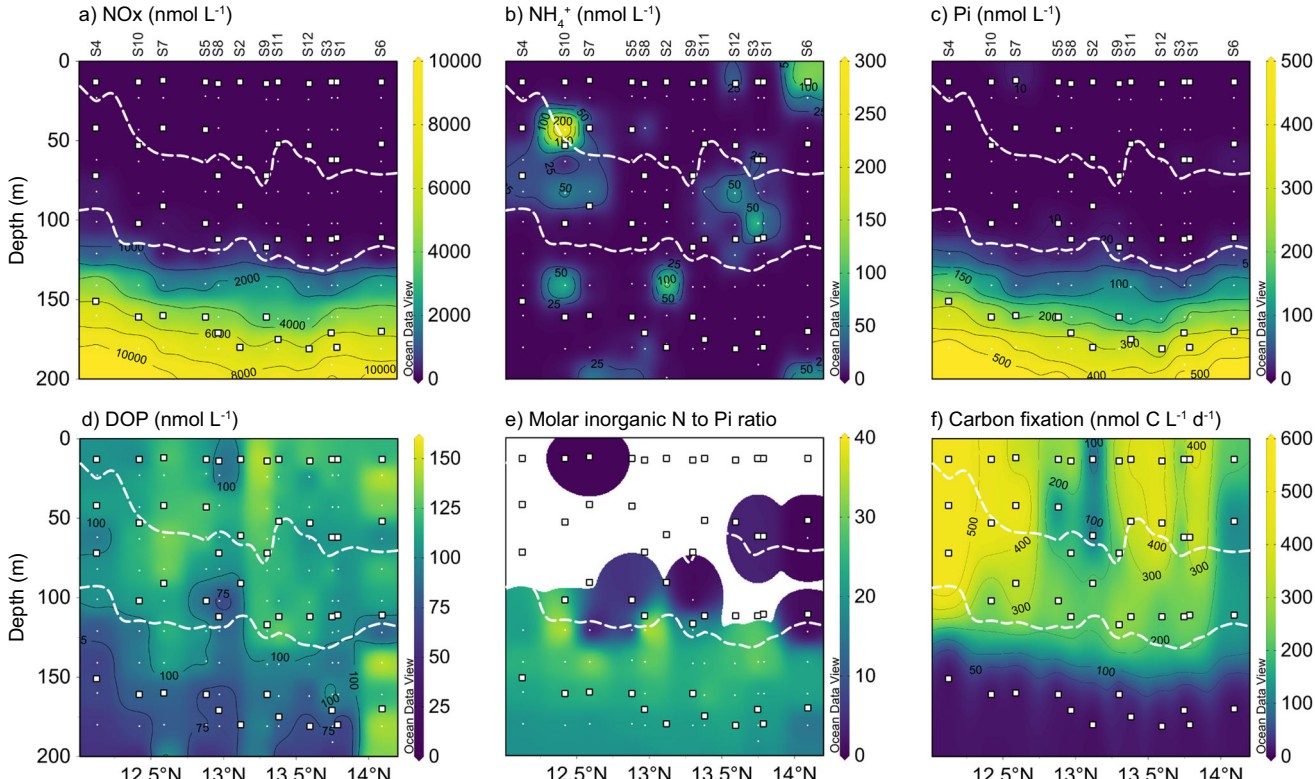

**Fig. 2 | Water column chemistry of nitrogen- and phosphorus-species and carbon fixation rates of the upper 200 m of the western tropical North Atlantic east of Barbados. a** Nitrate + nitrite (NOx). **b** Ammonium ($NH_4^+$). **c** Phosphate (Pi). **d** Dissolved organic carbon (DOP). **e** Molar inorganic nitrogen (NOx + ammonium) to phosphate ratios (N to Pi). **f** Carbon fixation rates. White squares represent depths of the incubation experiments and white circles indicate sampling depths for concentration measurements from the profiling CTD cast. The white dashed line represents the outline of the deep chlorophyll maximum. Lack of values in the N to Pi ratio panel is a consequence of largely non-detectable nitrogen and phosphate, as seen on (**a**–**c**). Source data are provided as a source data file.

surface and intermediate depths located above the DCM (Fig. 3), with median rates of 0.40 (0.21–1.31 interquartile range (IQR)) and 0.30 (0.17–1.36 IQR) nmol $CH_4$ $L^{-1}$ $d^{-1}$ respectively. The maximal rate of 9.57 nmol $CH_4$ $L^{-1}$ $d^{-1}$ was measured in the intermediate depth of station 4. The measured rates were comparable to rates reported from the surface waters of both subtropical gyres and the western North Atlantic[23,36] (0.08–4 nmol $CH_4$ $L^{-1}$ $d^{-1}$) even though direct comparisons are difficult due to different experimental designs. With increasing water depth, methane formation was still quantifiable but median rates decreased to 0.07 (0–0.25 IQR) and 0.06 (0–0.21 IQR) nmol $CH_4$ $L^{-1}$ $d^{-1}$ in the DCM and below respectively. Kruskal-Wallis statistical analysis revealed that methane formation was significantly different (H = 21, df = 3, $p < 0.01$) between the depths, except for the surface and intermediate depths (Pairwise Wilcoxon test, $p = 0.99$) and the DCM and below DCM depths (Pairwise Wilcoxon test, $p = 0.67$), separating the upper water column into an area of high methane formation above the DCM and an area of low formation at the DCM and below it. As our rates were mostly linear over 24 hours of incubation and often without a lag phase (Supplementary Fig. S4), the microbial community appears to be well adapted to the utilisation of methylphosphonate or other phosphonates.

Our data do not indicate that this process is turned on and off by a certain threshold Pi concentration, even though the highest rates from the methylphosphonate-supplemented experiment were consistently recorded in incubations with in-situ Pi concentrations below 25 nmol $L^{-1}$ (Supplementary Fig. S5). We also did not observe any clear correlation between the methane formation rates and the potential substrate availability (i.e. organic phosphorus concentration; Supplementary Fig. S5). This may be explained by the fact that

phosphonates only comprise a small part of DOP (5–10%)[30]. Cycling of only 0.25% of the phosphonate inventory at station ALOHA was suggested to be sufficient to explain the entire atmospheric flux of methane to the atmosphere there[17].

We should note that our measured rates should be treated as potential net rates, as the methylphosphonate additions (1 μmol $L^{-1}$) were in excess of the in-situ concentrations and potential methane consumption by methane oxidation could not be accurately quantified. Measurements of aerobic methane oxidation from open ocean surface waters are scarce and often in the range of tens of picomolar, even though occasionally rates of up to 4 nmol $L^{-1}$ $d^{-1}$ were measured[37]. Furthermore, anaerobic methane oxidation could be occurring in anoxic microniches, although all experiments were set up aerobically with ongoing photosynthesis. However, any methane oxidation occurring in our incubations would have ultimately led to an underestimation of our methane formation rates.

Very little is known about the microbial utilisation of methylphosphonate and the resulting methane formation in waters below the mixed layer. There are reports of increased carbon-phosphorus (C-P) lyase activity within the DCM in the subtropical North Pacific[38] and the highest methane concentrations in the subtropical North Atlantic were measured in the DCM[31] indicating aerobic methane formation below the mixed layer. This is peculiar because Pi is usually present below the DCM, and should be the preferred phosphorus source for microorganisms there[22]. Additionally, the C-P lyase is regulated by the phoregulon, and its expression is turned off in the presence of Pi[39]. In our incubations, methane formation remained low but quantifiable below the DCM (Fig. 3) despite the in-situ Pi concentrations of up to 384 nmol $L^{-1}$ (Fig. 2c).

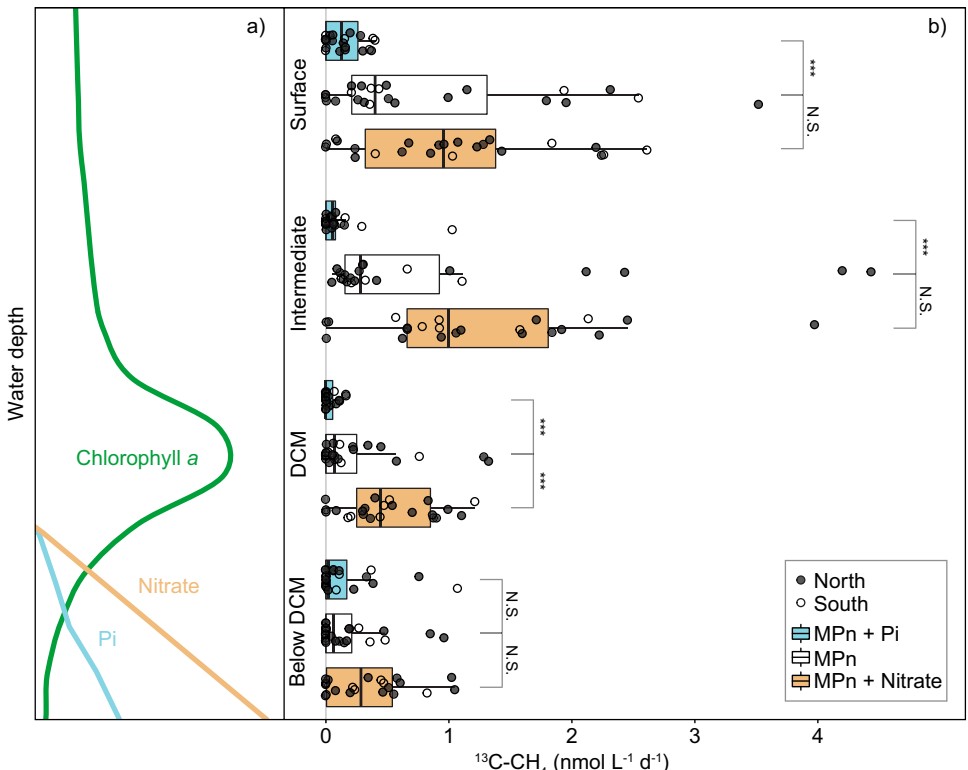

**Fig. 3 | Significant methane formation rates from $^{13}$C-methylphosphonate incubation experiments. a** Schematic drawing of a typical water column profile at the study site, showing the chlorophyll a (Chl a), as well as nitrate and phosphate (Pi) regimes. **b** Rate measurements of methane ($^{13}$C-CH$_4$) formation from the four incubation depths: 10-metre surface depth, intermediate depth, deep chlorophyll maximum (DCM) and below the DCM and the three incubation experiments: methylphosphonate-addition (MPn; white), methylphosphonate + phosphate-addition (MPn + Pi; blue) and methylphosphonate + nitrate-addition (MPn + nitrate; orange) from both duplicates and all twelve stations. Boxplots illustrate the 25–75% quantile range, encompassing the median rates with whiskers extending to data points within the 1.5 times the interquartile range. Each data point represents a

statistically significant (One-sided t-test, df = 2, p < 0.05, R$^2$ > 0.81) rate calculated from the linear regression of the first four time points (24 hours), with both insignificant and exponential rates reported as zero. Filled points represent the northern stations in the low surface Chl a waters and open points the southern stations in the high surface Chl a waters. Statistical difference (Pairwise Wilcoxon test, p < 0.05) between the MPn-addition and the others is highlighted (***p < 0.01, N.S. = not significant). From the intermediate depths, the two highest rates of methane formation are not shown in the plot, namely 4.8 and 9.6 nmol CH$_4$ L$^{-1}$ d$^{-1}$ from S3 and S4 of the MPn-addition experiment respectively, as well as the 9.4 nmol CH$_4$ L$^{-1}$ d$^{-1}$ from the MPn + nitrate-addition experiment from S12. Source data are provided as a source data file.

## Effect of inorganic nutrients on methane formation

This pervasive utilisation of methylphosphonate in the presence of Pi was confirmed by our methylphosphonate + Pi-addition experiment (Fig. 3) where methane formation rates were significantly (Pairwise Wilcoxon test, p < 0.05) repressed−but not completely inhibited−in the upper three incubation depths, compared to the methylphosphonate-addition incubations. It therefore seems that while elevated Pi concentrations tend to repress aerobic methane formation in general, some methylphosphonate utilisation persists even in the presence of Pi (Supplementary Note 3). It should be kept in mind that in our experiments, methylphosphonate and Pi were added in equimolar amounts, as previous observations from cultures[40,41] and field experiments[42] suggest that the methylphosphonate:Pi ratio might affect the degree to which methylphosphonate utilisation is inhibited. However, even under equimolar additions the degree of repression varies strongly between organisms (e.g. 98% in cultured *SAR11 str HTCC7211* vs. 70% repression in *Trichodesmium*[40,41]). Interestingly, in depths below the DCM, additions of Pi did not significantly repress methane formation from methylphosphonate (Fig. 3). As both carbon fixation and POC concentrations were lowest in this depth (Supplementary Fig. S5), it is feasible that availability of labile DOM is an additional controlling factor that affects the rate of methylphosphonate utilisation in depths, where dominant methylphosphonate utilisers are predominantly mixo- or hetero-trophic. The important implication of these observations is that the utilisation of

methylphosphonate may extend into environments that are less chronically Pi-limited, thus expanding the potential range for aerobic methane formation.

In contrast to Pi, additions of nitrate were expected to increase the rates of methylphosphonate-dependent methane formation, especially as inorganic nitrogen was also limiting in the upper water column (Fig. 2). Addition of nitrogen have been shown to enhance both primary productivity[43] as well as heterotrophic productivity[28] in the Atlantic; moreover, nitrogen supplementation was expected to exacerbate phosphorus stress and thus create even more favourable conditions for the utilisation of methylphosphonate. Nitrogen addition experiments have been shown previously to lead to increased methane production rates in the subtropical North Pacific[10,22]. Correspondingly, the median rates of methane formation in the methylphosphonate + nitrate-addition incubations increased substantially across all four depths, even though the difference to the methylphosphonate-addition experiment was not statistically significant for three out of four depths (Pairwise Wilcoxon test, p > 0.05), the exception being the DCM (Pairwise Wilcoxon test, p = 0.01). This may be due to the high heterogeneity of the rates observed in the upper two incubation depths. The median rate increase of methane formation (by 2.4–6.3) in our methylphosphonate + nitrate-addition experiment (Fig. 3) was comparable to the previously observed increase of 1.7–2.8[22]. We should note that nitrate additions could have simultaneously enhanced methane oxidation rates, as many

methanotrophic bacteria and archaea can use nitrate as an electron acceptor. Increased rates of methane oxidation would have obscured the true impact of nitrate on methane production.

### C-P lyase enzyme mainly encoded by *Trichodesmium* and *Alphaproteobacteria*

The observed decline of methylphosphonate-driven methane formation rates with depth was accompanied by a decrease in the relative abundance of *phnJ*-containing microorganisms in each size fraction [large (>10 μm), medium (3–10 μm) and small (0.22–3 μm); Table 1]. In the surface and intermediate depths, 14.7 and 4.2% of all microorganisms in the large size fraction contained the *phnJ* gene respectively, compared to 2.5 and 2.2% in the small one. In the medium size fraction, only around 1% of all microorganisms from the upper two depths contained the *phnJ* gene, indicating that comparatively this size

fraction is least important. At the DCM, less than 1% and below the DCM, less than 0.1% of all microorganisms in each size fraction encoded for the *phnJ* gene. The only exception to this was the large fraction in the below DCM depth, where the relative abundance was still 4.7%. A similar pattern has been observed at the Bermuda Atlantic Time Series station, with the *phnJ* containing community decreasing ca. 2–3-fold between 20 and 100 m depth[44].

Within the large size fraction, the *phnJ* genes were taxonomically affiliated almost exclusively with the filamentous cyanobacterial genus *Trichodesmium* (Fig. 4). Their distribution matched the percentage of microorganisms from this size fraction containing the *phnJ* (Table 1), suggesting that *Trichodesmium* was the dominant *phnJ* containing organism in the large size fraction. Additionally, some of the *phnJ* containing bacteria identified in the larger size fraction may constitute part of the *Trichodesmium* microbiome in-situ, as previously shown[45]. While most abundant above the DCM, the *Trichodesmium phnJ* was still detected in ca. 4% of the genomes in the large size fraction below the DCM (Table 1), possibly due to migrating or sinking cells[46]. *Trichodesmium* is a key primary producer, as well as a nitrogen-fixing organism in the western North Atlantic[47] and has the capacity to utilise methylphosphonate and release methane even in the presence of Pi[10,40,48]. Moreover, some *Trichodesmium* species also have the capacity to synthesise phosphonates[49], albeit not methylphosphonate specifically[50], and may thus contribute to phosphonate cycling at this site (see also Supplementary Note 4 and 5).

Our combined results indicate that *Trichodesmium* may contribute to methylphosphonate utilisation in the surface and intermediate depths during both Pi-deplete and replete conditions. Due to their large size and patchy distribution, *Trichodesmium* may also

**Table 1 | Relative abundance of the *phnJ* gene from the different size fractions of station S5 normalised to the *recA* gene, assuming a single copy per bacterium**

| Depth | Small (0.22–3 μm) | Medium (3–10 μm) | Large (>10 μm) |
|---|---|---|---|
| Surface | 2.47% | 1.33% | 14.71% |
| Intermediate | 2.24% | 1% | 4.18% |
| Deep chlorophyll maximum | 0.8% | 0.16% | 0.56% |
| Below deep chlorophyll maximum | 0.04% | 0.04% | 4.66% |

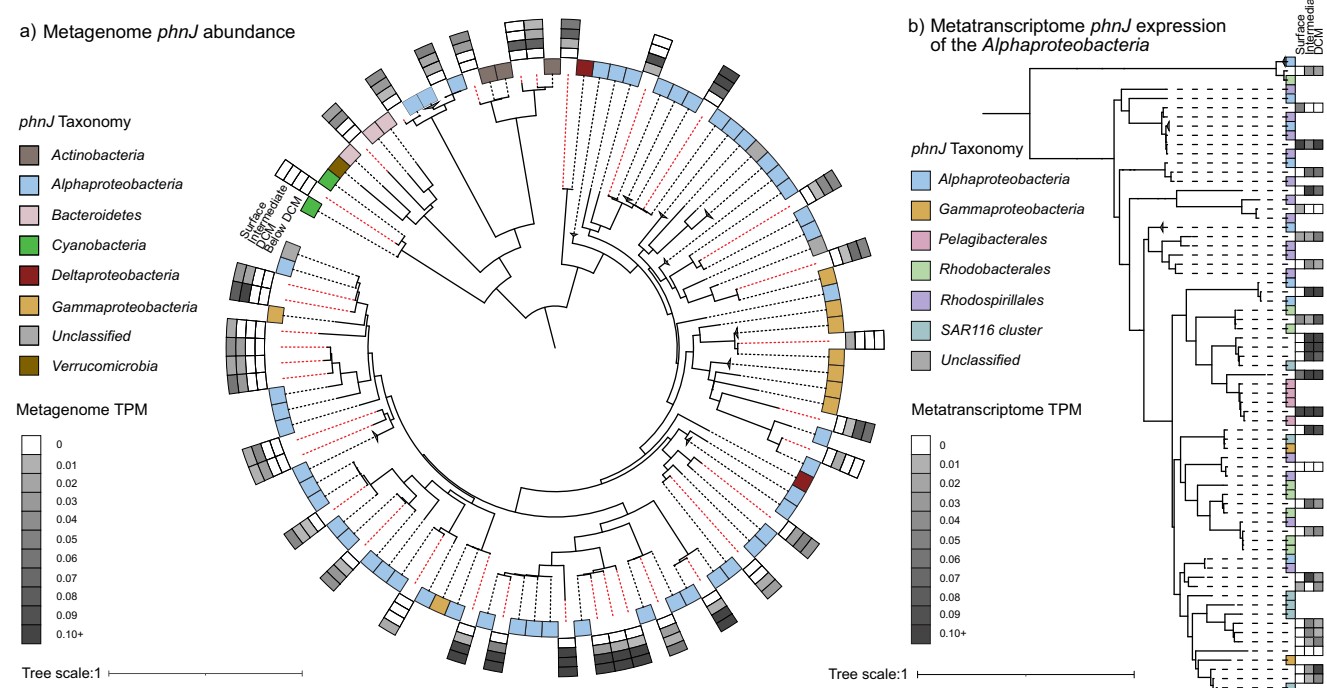

**Fig. 4 | *PhnJ* gene diversity, relative abundance and expression. a** Phylogenetic tree of all assembled *phnJ* gene sequences from all depths (10-metre surface depth, intermediate depth, deep chlorophyll maximum (DCM) and below the DCM) of the smallest size fractions and the respective reference sequences. The NCBI taxonomy of the reference sequences is indicated by the colour code of the inner boxes, the new sequences are indicated by the dashed red line. Relative abundance (TPM values) of the *phnJ* gene sequences in the metagenomic samples from the small (0.22–3 μm) size fraction in the four different depths are shown in the outer four boxes. In the metagenomes of the large (>10 μm) size fraction, only the sequence clustering with *Cyanobacteria* showed high TPM values in the surface, intermediate and DCM depths (0.77, 0.18 and 0.13 respectively; data not shown in the figure). **b** Pruned phylogenetic tree of *phnJ* gene sequences clustering with *Alphaproteobacteria*. NCBI taxonomy of the reference sequences is represented by the colour code in the first column. Transcription levels (TPM values) in the metatranscriptome samples of the small (0.22–3 μm) size fraction of the upper three depths are depicted in the following three columns. For more information, refer to Supplementary Fig. S6. Source data are provided as a source data file.

contribute to the high heterogeneity of the measured methane formation rates (a difference of up to 8.5 nmol $CH_4$ $L^{-1}$ $d^{-1}$ between duplicates in the intermediate depth of S4). This heterogeneity was often more pronounced above the DCM, supporting the presumed key role of larger phytoplankton and/or particle-associated bacterioplankton in methylphosphonate-driven methane formation[22,48].

Within the small size fraction, the *Alphaproteobacteria* were the most numerous, and their overall abundance matched the observed depth distribution of the *phnJ*-encoding microorganisms of this fraction in the upper three incubation depths (Fig. 4). The *phnJ*-containing *Alphaproteobacteria* were very diverse, with sequences clustering with representatives of *SAR116*, *Pelagibacterales*, *Rhodobacteraceae* and *Rhodospirillaceae* (Supplementary Fig. S6), similar to previously observed C-P lyase containing microbial communities in e.g. the Sargasso Sea[18,48]. *Pelagibacterales* have been shown to produce methane from methylphosphonate under Pi-limitation in culture experiments[41]. Due to their high abundance, particularly in regions like the Sargasso Sea where they can make up 52% of the C-P lyase-containing community[44], they are expected to be important global contributors to methylphosphonate-driven methane formation. *SAR116* is also known to utilise phosphonates under Pi-limitation[51] and has been shown to be a dominant group in both the Sargasso- and the Mediterranean Sea[18]. The *Rhodobacterales* are a bacterial taxon that includes competitive opportunists with the capacity to form blooms[52], that were shown to increase in abundance after methylphosphonate addition[36]. As such, they too may have contributed to the observed exponential nature of some of the methane formation rates.

Transcriptomic analysis of the small fraction revealed in-situ transcription of the C-P lyase pathway (Fig. 4b), with most *phnJ* transcripts assigned to *Alphaproteobacteria* followed by *Gammaproteobacteria* (Supplementary Fig. S6). Interestingly, relative transcript abundances did not follow the vertical rate distribution, with higher C-P lyase transcription in the intermediate and DCM depths compared to the surface. This suggests that comparatively, alphaproteobacterial phosphonate utilisation becomes more important deeper in the water column, whereas the large size organisms like the nitrogen-fixing cyanobacterium *Trichodesmium* appear to dominate above the DCM. The widespread distribution and expression of the *phnJ* gene across many microbial taxa in-situ (Fig. 4) combined with the immediate onset and linear trend of the methane formation rates strongly implies that the microbial community was well adapted to metabolise methylphosphonate in particular and phosphonates in general through the C-P lyase pathway. As the *phnJ*-containing taxa include some of the most abundant marine auto- and heterotrophic microorganisms, methylphosphonate or more general phosphonate utilisation might be a key metabolic feature for microorganisms living in Pi-limited oligotrophic oceanic regions where primary productivity relies on recycled nutrients.

## Aerobic methane formation and phytoplankton activity

Aerobic methane formation in aquatic environments (both marine and freshwater) is linked to the activity of primary producers[1,53], either through direct methane formation from DIC during carbon fixation[15,16] or through the metabolism of methylated compounds such as methylphosphonate[10,22]. At our study site, the processes of aerobic methane formation and primary production can be linked through methylphosphonate-utilisation by the primary producer *Trichodesmium*. Despite this potential direct link, neither the surface Chl *a* concentrations, the parameter used to estimate primary productivity, nor our directly measured carbon fixation rates, showed a significant correlation with methane formation rates (Supplementary Figs. S5, S7). Hence, our results suggest that surface Chl *a* and carbon fixation rates may not be universally suitable as predictors of aerobic methane formation. We also investigated the possibility that DIC can be converted to methane in the investigated study site, for which we set up parallel

experiments in duplicate using only $^{13}C$-DIC. However, out of 96 incubations we were only able to detect one substantial rate (0.58 nmol $CH_4$ $L^{-1}$ $d^{-1}$), in the DCM of station S12. Moreover, due to the long incubation time (48 hours) we cannot say with confidence whether methane was produced directly from DIC, or from any of the many carbon intermediates that are formed during autotrophic cell metabolism. We conclude that DIC likely was not a major contributor to the methane formation in the western tropical North Atlantic at this time, with any rates being either hugely sporadic or much lower than the ones from methylphosphonate. Instead, demethylation of methylphosphonate by phytoplankton and bacterioplankton was likely the main source of aerobic methane formed in this region.

## Role of methylphosphonate in carbon drawdown

In Pi-limited regions, organic phosphorus compounds represent a key phosphorus source to sustain productivity[54]. For example, DOP degradation by alkaline phosphatases could explain 12–30% of the phosphorus requirements in the North Atlantic subtropical gyre[55] but similar estimates are lacking for phosphonate degradation through C-P lyases. As methane and Pi are stoichiometrically formed during methylphosphonate demethylation[19], our measured rates of potential net methane formation can be used to determine the maximum of Pi liberated during this process to potentially fuel e.g. primary production. Based on the median net carbon fixation rate in the surface depth (371 nmol C $L^{-1}$ $d^{-1}$), we estimate a redfieldian Pi requirement of 3.5 nmol $L^{-1}$ $d^{-1}$. Considering the range of our measured methane formation rates, (methyl)phosphonates could support 0–100.7% (median 11%) of the Pi needed to sustain the measured rates of primary production, thus making them an important but so far poorly understood contributor of the contemporary oceanic carbon drawdown.

Gross primary productivity in the modern open ocean is sustained to 90% by recycled phosphorus[54]. Under future climate projections, warmer surface waters and the resulting enhanced stratification will reduce mixing of the nutrient rich deeper waters into the surface, likely decreasing primary productivity in the long run[56,57]. Under these conditions, utilisation of organic phosphorus (as well as nitrogen) may become comparably more widespread, as indicated e.g. by the prevalence of the C-P lyase pathway in extremely Pi-limited environments[18,58]. Phosphonates are ubiquitous in surface marine waters and they have been shown to be rapidly turned over (on the time scale of a few days) due to an intense redox cycling between +5 and +3 oxidation states[30]. If the recycling of phosphorus in the surface waters of the nutrient limited regions intensifies, each phosphorus atom may undergo more cycles of methylation (or alkylation, in general) before being removed from the mixed layer. Combined with the projected expansion of Pi-limited oceanic regions, methylphosphonate-driven aerobic methane formation is likely to increase in the long run. However, increasing stratification is also predicted to favour smaller picocyanobacteria[59] like *Prochlorococcus*, which has been shown to encode an alternative methylphosphonate degradation pathway that leads to the formation of formate instead of methane[60]. To realistically assess the interplay of those two scenarios and its impact on open-ocean methane emissions, it is imperative to better constrain microbial formation and utilisation of methylphosphonate above and below the mixed layer.

## Methods

### Sample collection

During the R/V Meteor cruise M161, samples were collected between 19 January and 20 February 2020 from twelve stations in the western Tropical North Atlantic off Barbados (Fig. 1). Authorisation to conduct marine scientific research in territorial waters of Barbados was granted by the Ministry of Foreign Affairs and Foreign Trade, Barbados (permit no. IR/2020/08 issued on 15 January 2020). At each

station, hydrographical data was recorded by a Sea-Bird conductivity temperature depth (CTD) system equipped with sensors for temperature and salinity (Sea-Bird), dissolved oxygen (Sea-Bird 43), fluorescence (WETLabs) and turbidity (WETLabs), mounted on a 24-way stainless steel frame system equipped with 10 litre Niskin bottles. Water samples were collected from two subsequent CTD casts at each station between 12:30 am and 4:00 am local time to start incubations at sunrise. The first cast profiled the upper 800 m, with discrete water samples taken at regular intervals in the upper 200 m and a reference sample from around 250 m. From each depth, samples were collected for nutrients (nitrate ($NO_3^-$), nitrite ($NO_2^-$), ammonium, phosphate (Pi) and total phosphorus (TP)), methane, DNA and RNA, chlorophyll $a$ (Chl $a$), particulate organic -carbon (POC), -nitrogen (PON) and -phosphorus (POP). On the second cast, four discrete depths in the upper 200 m were selected based on chlorophyll concentrations: below the deep chlorophyll maximum (DCM), at the DCM, an intermediate depth above the DCM and a 10-metre surface depth. The 10-metre surface depth was chosen due to technical restrictions of sampling with a CTD rosette and to ensure a comparable depth within the surface mixed layer (Supplementary Fig. S1). From these depths, samples were taken for nutrients, methane, POC, PON, DNA and RNA, as well as water samples for stable isotope incubation experiments to quantify rates of aerobic methane formation and carbon fixation.

## Water column chemistry

Ammonium concentrations were determined fluorometrically (orthophthaldialdehyde (OPA) method[61]) using unfiltered water samples immediately after sample retrieval. The limit of detection (LOD), calculated as the mean of the blanks plus three times the standard deviation, was 24 nmol $L^{-1}$. Samples for $NO_3^-$ and $NO_2^-$ were frozen unfiltered at −20 °C until simultaneous spectrophotometric measurement using a QuAAtro39 autoanalyser (Seal Analytical) using standard methods[62]. It should be noted that freeze thawing may break open cells and release intracellular nutrients[63]. $NO_3^-$ was determined as the difference between $NO_3^-$ and $NO_2^-$ combined (NOx) and $NO_2^-$. LODs were 583 nmol $L^{-1}$ (NOx) and 33 nmol $L^{-1}$ ($NO_2^-$). From the same sample inorganic phosphate (Pi) and total phosphorus (TP) were also determined. These were measured simultaneously using a Liquid Waveguide Capillary Cell (Seal Analytical). Briefly, for TP measurements, TP was irradiated in a UV digester after persulfate addition; this is followed by acid hydrolysis at 90 °C before colorimetric measurement at 880 nm after reacting with molybdate, antimony and ascorbic acid (Seal Analytical). LODs for Pi and TP were 10 nmol $L^{-1}$ and 30–50 nmol $L^{-1}$, respectively. The organic phosphorus (OP) fraction was calculated by subtracting Pi from TP.

Samples for POP were collected by filtering 3.5 L of seawater onto pre-combusted (4 hours at 450 °C) glass fibre (GF/F, Whatman) filters. POP was measured using a modified version of the persulfate oxidation method[64]. Briefly, instead of autoclaving, the samples were heated in a microwave "Mars 1" (CEM) for 20 minutes to reach 150 °C and were then incubated for 15 minutes, followed by 20 minutes of cooling. Subtracting the POP from the calculated OP (see previous section), yielded the dissolved organic phosphorus (DOP) concentrations. POC and PON were determined from an additional 3.5 L of seawater filtered onto pre-combusted GF/F filters (Whatman) and dried at 50 °C overnight. Samples were decalcified by hydrochloric acid (37%, Merck) fumes in a desiccator overnight, dried and pelleted in tin cups and subsequently measured by an element analyser (Thermo Flash EA, 1112 Series) coupled to a continuous-flow isotope ratio mass spectrometer (Delta Plus XP IRMS, Thermo Finnigan).

Duplicate samples for methane concentrations were filled headspace free with a triple overflow into 60 mL serum bottles using gas-tight Viton tubing. Microbial activity was inhibited by adding a scoop

of copper(I)chloride (Merck) and the bottles were stoppered (Glasgerätebau Ochs) and crimped. Before analysis, a 20 mL headspace was set with synthetic air (80% nitrogen, 20% oxygen; Air Liquide) before methane measurements on a gas concentration analyser (Picarro G2308). Samples were calibrated against three standards: degassed MiliQ water, air-saturated MiliQ water and air-saturated saline solution (10% sodium chloride). However, an interference from the stoppers (Butyl-Septen für Hals N20, Glasgerätebau Ochs Laborfachhandel) through apparent methane leakage invalidated the measurements, making them void. This was confirmed by observing increases in methane concentration over time in a MiliQ experiment that was initially air saturated.

For discrete water column Chl $a$ samples, 600 mL of water was filtered onto a 25 mm pre-combusted GF/F filter (Whatman). Chl $a$ was extracted in 90% acetone from filters and measured fluorometrically (Turner Designs[65]). Results were calibrated against a Chl $a$ standard (Sigma Aldrich). The concentration measurements from all depths and stations were then used to calibrate the fluorescence sensor of the CTD, by linear regression, converting relative fluorescence units into µg $L^{-1}$ Chl $a$ (Supplementary Fig. S8) for stations 3 and 7, where Chl $a$ samples were not taken. Satellite surface Chl $a$ concentrations were obtained from Earth Data and downloaded through the Giovanni online data system (https://giovanni.gsfc.nasa.gov/giovanni/)[66], which is developed and maintained by the NASA Goddard Earth Sciences Data and Information Services Centre (GES DISC), for the timeframe between 01 January and 29 February, creating a composite snapshot using the MODIS-Aqua, monthly, 4 km resolution[67] for the region between 56.75–60° W and 11.75–15° N.

## Methane formation rate measurements and calculations

Rates of methane formation were quantified at each station from four depths from $^{13}C$ stable isotope incubation experiments with either dissolved methylphosphonate ($^{13}CH_5O_3P$, 99 atom % $^{13}C$, dissolved in MilliQ water, Sigma-Aldrich) or dissolved inorganic carbon (DIC; $NaH^{13}CO_3$, 98 atom % $^{13}C$, dissolved in MilliQ water; Sigma-Aldrich) set up in biological duplicates. Briefly, acid-washed 250 mL serum bottles were filled headspace-free, stoppered (Geo-Microbial Technologies) and crimped. A 50 mL headspace was set with synthetic air (80% nitrogen, 20% oxygen; Air Liquide) before tracers were added. All bottles were swirled for mixing after the additions. Four experiments were set up in separate incubation bottles: methylphosphonate-addition (1 µmol $L^{-1}$ $^{13}C$-methylphosphonate), methylphosphonate + Pi-addition (1 µmol $L^{-1}$ $^{13}C$-methylphosphonate + 1 µmol $L^{-1}$ Pi), methylphosphonate + nitrate addition (1 µmol $L^{-1}$ $^{13}C$-methylphosphonate + 16 µmol $L^{-1}$ $NO_3^-$) and the DIC incubation (200 µmol $L^{-1}$ $^{13}C$-DIC). In one replicate of both the methylphosphonate-addition and DIC incubation experiments 10% of the water was replaced by deuterium oxide ($D_2O$, 99.9 atom %, Sigma Aldrich). The incubation bottles were incubated in two *on-deck* incubators at surface water temperatures and close to in-situ light conditions. To achieve this, the incubators were covered with two different blue filters (Ocean Blue #724 and Tokyo blue #071, Lee Filters https://www.leefilters.com/lighting/colour-list.html), with the below DCM and DCM as well as the intermediate and surface depths incubated together. At five time points, after 0, 6, 12, 24 and 48 hours of incubation, the headspace was subsampled. At each time point, 5 mL of the headspace was sampled with a gas-tight syringe whilst replacing the removed volume with synthetic air. The headspace sample was injected into a 12 mL Exetainer (Labco UK) pre-filled with MilliQ-water, setting a 5 mL headspace for later analysis. Gas samples were preserved through the addition of 100 µL mercuric chloride solution ($HgCl_2$, 0.7 g/100 mL) to inhibit microbial activity and stored overhead. Samples were measured on a cavity-ring down spectrometer (G2201-i coupled to Liaison A0301, Picarro Inc., connected to an AutoMate Prep device, Bushnell).

Methane formation rates were calculated from incubations that showed a statistically significant linear increase of excess $^{13}$C-methane over the first four time points (24 hours; Supplementary Fig. S4). Significance was determined by a one-sided student's $t$-test ($p < 0.05$) applied to the linear regression ($R^2 > 0.81$) of the excess $^{13}$C-methane formation. Clearly exponential rates (formation only in the 24 hour time point) were set to zero to ensure comparable analysis, this only affected a single incubation. All rates were corrected for the dilution factor and labelling percentage. For methylphosphonate-amended incubations, the labelling was assumed to be 100%, as the in-situ concentrations cannot be determined. For the DIC-amended incubations, labelling was ~10%. To compare the methane formation rates across depth and incubation type, Kruskal-Wallis statistical analysis was conducted after a Shapiro-Wilk normality test followed by a pairwise Wilcoxon test with a Benjamini-Hochberg adjusted $p$-value, all one-sided and performed in RStudio. To investigate correlating parameters with methane formation rates, Spearman Ranked correlation analysis was performed in RStudio using all available parameters.

## Carbon fixation rates
Carbon fixation experiments were set up in parallel bottles to the aerobic methane formation experiments. Briefly, biological triplicate water samples in 4.6 L polycarbonate bottles were amended with $^{13}$C-DIC (NaHCO$_3$, >98% at% $^{13}$C, Sigma Aldrich) to achieve a labelling percentage of ~5 at% and gently agitated to mix for 15 minutes. Bottles were then incubated, headspace-free, for 24 hours under dawn to dawn conditions in the two *on-deck* incubators (see above). One additional unamended sample per depth was also incubated as a natural abundance control incubation. After 24 hours, 3.5 L were filtered onto a pre-combusted (4 hours at 450 °C) 25 mm GF/F filter (Whatman) and dried at ~55 °C overnight. Filters were processed as described above (POC/PON) and samples were measured with an element analyser (Thermo Flash EA, 1112 Series) coupled to a continuous-flow isotope ratio mass spectrometer (Delta Plus XP IRMS, Thermo Finnigan). Rates of primary productivity were determined from the uptake of $^{13}$C-DIC into biomass[68] with carbon fixation rates calculated according to Großkopf[69].

## DNA and RNA isolation and metagenome and metatranscriptome sequencing
At each incubation depth, twice ten litres of seawater were sequentially filtered through a 10 μm pore size polycarbonate filter and a 3 μm pore size polycarbonate filter before being split into two Sterivex filters (0.22 μm pore size, each received 5 L of water). All filters were immediately frozen in liquid nitrogen and stored at −80 °C until DNA or RNA extraction. Station 5 was chosen as a representative station for DNA and RNA extraction. DNA was isolated by the chloroform/isoamyl alcohol (24:1) method[70]. RNA was extracted by standard protocol of the RNeasy PowerWater kit (Qiagen), with an additional heating step at 65 °C for 10 minutes prior to bead beating. Samples were sequenced with 2 * 150 bp paired-end Illumina technology. RNA samples were sequenced after rRNA depletion, which was only possible for the small (0.22–3 μm) size fraction of the upper three incubation depths due to limited RNA quantities. For metagenomes a total of >10 Gb and for metatranscriptomes a total of >20 Gb were sequenced per sample.

## Metagenomic and Metatranscriptomic analysis
Metagenomic and metatranscriptomic reads were adapter and quality trimmed using Trimmomatics version 0.39[71]. Metagenomic samples were used for distance estimation and clustering using mash version 2.3 (Supplementary Fig. S10;[72]), revealing that the surface and intermediate depths as well as the DCM and below DCM samples and the different size fractions, clustered together. Based on these clusters, the samples were co-assembled using megahit version 1.2.9[73] for a total of six co-assemblies. All co-assemblies were then put into the

SqueezeMeta pipeline version 1.3[74]. We used the default prodigal setting with the –meta option, which resulted in the translation of sequence OQ435767 using translation table four. From all open reading frames, 135 genes annotated as *phnJ* were extracted. The amino acid sequences were clustered with CD-HIT version 4.6[75–78], using a 95% sequence identity cut-off over 90% of the length of all sequences, resulting in 78 unique sequences. The sequences were used in blastp search[79] to the NCBI nr database (2$^{nd}$ December 2021). The best two hits per query were used as references, resulting in 104 unique reference sequences. A multiple sequence alignment was conducted with MAFFT v7.475[80], manually filtered and trimmed with trimAL v1.4.rev15[81]. This left 128 sequences, 36 from the original samples, to construct a phylogenetic tree with IQ tree including the Modelfinder[82–84]. To map both the metagenomic and metatranscriptomic reads to the tree sequences, bowtie2 version 2.3.4.1[85] was used. Read counts with a sequence identity >80% were extracted using coverM version 0.6.1. TPM values for both metagenomic and metatranscriptomic samples were calculated in accordance with Wagner[86].

To constrain the relative *phnJ* abundance within the different size fractions, *phnJ* read counts were corrected for read counts of the single copy marker gene *recA*[44]. *RecA* sequences were identified in the ORF table and clustered at 95% sequence identity over at least 90% sequence length using CD-HIT version 4.6[75–78]. Metagenome reads were mapped onto the clusters with bowtie2 version 2.3.4.1[85]. Hits were then filtered based on the sequence identity of >80% using coverM version 0.6.1. Read counts of both *phnJ* and *recA* genes were normalised by ORF length and set in relation.

## Figure generation
Maps and nutrient figures were generated using the quantum geographic information system (QGIS; version 3.18.3) or Ocean Data View (version 5.1.7[87]). Rate analysis was conducted in Microsoft Excel (2016) and figures were generated in RStudio 2023.3.0.386[88] versions 4.1.3 and 4.2.2 (R Core Team) using the ggplot2[89] package. The phylogenetic tree was visualised in iTOL[90]. All figures were processed in Adobe Illustrator.

## Reporting summary
Further information on research design is available in the Nature Portfolio Reporting Summary linked to this article.

## Data availability
The *phnJ* sequences generated in this study have been deposited in the NCBI database under accession codes OQ435733–OQ435768. See Source Data for Supplementary Fig. S6 for a full list of accession codes, including those for reference sequences. Satellite surface Chl *a* concentrations were obtained from Earth Data using MODIS-Aqua monthly 4 km resolution[67] and downloaded through the Giovanni online data system (https://giovanni.gsfc.nasa.gov/giovanni/)[66], which is developed and maintained by the NASA GES DISC. Source data are provided with this paper.

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

## Acknowledgements

We would like to thank the Leitstelle of the DFG, captain and crew of the FS Meteor during the M161 cruise for all their assistance during the expedition; D. Baranowski and his CTD team for helping with the sampling, providing the CTD data and calculating the mixed layer depths; B. Stevens and S. Bony for coordinating the EUREC⁴A project; the MODIS

mission scientists and associated NASA personnel for the production of the data used in this research effort; M. Knutzen, G. Klockgether, S. Lilienthal, K. Imhoff, S. Piosek, N. Rujanski and D. Tienken for the technical support; S. Ahmerkamp, C. Frey, F. M. Jalaluddin, K. Kitzinger, H. Marchant, A. S. McDermott and D. Speth for the helpful discussions and comments on the manuscript.

The M161 cruise was financially supported by the Deutsche Forschungsgemeinschaft (Fördernummer: GPF18-1-69, EUREC$^4$A$^{++}$), Bundesministerium für Bildung und Forschung. This study was financially supported by the Max Planck Society.

## Author contributions

W.M., A.T.K. and J.v.A. conducted the sampling. J.v.A. performed the incubation experiments for methane formation and measured the samples. Rate analysis and calculations were conducted by J.v.A. and G.L. Carbon fixation experiments were conducted by A.T.K. and W.M. DNA and RNA was extracted by J.v.A. and metagenomic and metatranscriptomic analysis was performed by M.P. S.S. and M.M.M.K. assisted with study design and data analyses, and J.v.A and J.M. wrote the manuscript with contributions from all other authors.

## Funding

## Competing interests

The authors declare no competing interests.
