## [Peer Review File · Nature Communications]

Methylphosphonate-driven methane formation and its link to primary production in the oligotrophic North AtlanticREVIEWER COMMENTS

Reviewer #1 (Remarks to the Author):

The study on methylphosphonate-driven methane formation by Arx et al. presented here is an interesting field study addressing which abiotic parameters and present bacterioplankton matter for methane formation. The results are especially needed to understand and predict better biogeochemical cycling in changing water bodies. In general, the manuscript is well-written and well-structured. However, I am unsure if it matches the broad readership of Nature Communications as it is quite detailed and field-specific (based on one field sampling campaign). The study might benefit from comparing two distinct field studies via, e.g. adding data from a field sampling elsewhere. The work will be mainly of interest to readers in the field of marine biogeochemistry and closely-related fields such as limnology and atmospheric sciences. I think the manuscript can benefit from including a few more explanatory sentences here and there. Concerning the methodology, I think apart from the molecular measurements (not my field of expertise), all used methods and analyses are sound and well-described in enough detail. I have one major issue, and a few minor issues outlined below. Otherwise, I enjoyed reading this manuscript, and hope to see it published soon.

Major comment:

L. 69-71: A particulate organic N:P ratio of 20 indicates merely P-limitation, and absolutely not a strong P-limitation. See e.g. Ptacnik et al. (2010). I recommend revising this part and the discussion accordingly.

Minor comments:

L. 17: Replace “such as” with e.g. including or most dominantly.

L: 69-71: Reference for the Redfield ratio is needed, i.e., Redfield (1958), to aid the broad readership of Nature Communications.

L. 74: Please consider including the explanation here why the C-fixation rates declined 100-fold below the DCM.

L. 127-128: Shorting header to increase readability.

L. 241-245: Please explain why a 10-m surface depth was selected, as often 1-m surface

depth is used.

Caption Fig. 1c: Please state that you determined the molar N:P ratio here.

L. 606-610: Please consider including a reference for RStudio software here for consistency.

Ptacnik, R., Andersen, T., & Tamminen, T. (2010). Performance of the Redfield ratio and a family of nutrient limitation indicators as thresholds for phytoplankton N vs. P limitation. *Ecosystems*, 13, 1201-1214.

Reviewer #2 (Remarks to the Author):

This study investigates the formation of methane in the eastern tropical/subtropical Atlantic Ocean, a region that is strongly phosphate limited and by inference a site where methane production from methylphosphonate is to be expected.

The study makes several original contributions that are noteworthy: 1) the study provides the first potential net rate estimates of methane formation through the methylphosphonate (MPhn) pathway for natural communities of marine microbes. 2) the study demonstrates that MPhn is metabolized to methane even under relatively high (> 100 nM) phosphate concentrations supporting genomic and C-P lyase rate measurements that methane formation occurs in deeper, phosphate rich waters below the euphotic zone, 3) the study highlights a possible connection between nitrogen fixing *Trichodesmium* (or at least its microbiome) and methane formation, and 4) the study finds little evidence for direct production of methane from primary producers, suggesting at least in this limited study and at least in P-limited regions, direct production might not be that important a pathway.

I found the manuscript to contribute original results to the topic of aerobic methane production in the ocean, which will influence and be influenced by climate change. The topic is a relatively new line of research in the field, and I therefore think the paper will be of broad interest to the community. The work was carefully done and the results and conclusions were sound.

Nevertheless, I felt the paper could be substantially improved with moderate revision by the

authors. I think the paper would be more interesting and stimulating if the authors dug a bit deeper into the questions addressed and by placing the data here within the data and conclusions reached by others (aside from a mere citation). Much of the discussion in the paper follows issues outlined in previous papers. The new measurements made by the authors offers a chance to more fully examine or discuss some of these questions, and I felt the authors missed an opportunity to do so. For example, Holmes et al. GBC 2000 made estimates of methane flux to the atmosphere in the Sargasso Sea. Assuming (as Holmes et al does) that all of the methane produced in the euphotic zone is ventilated, at steady state how do your methane production values compare with Holmes et al.'s efflux data? There are several possible outcomes and interpretations if the two sets of measurements align or not.

The authors point out that phosphonate degradation is regulated by the concentration of phosphate although the details are not known. Given your new data, there appears to be more you can say about this. As outlined on page 4, other studies have shown that likely rates of C-P lyase activity are high in and below the DCM, and genomic studies suggest the presence of C-P lyase (and other phosphonate utilization pathway) genes and C-P lyase activity into the deep ocean. The repression in rate found here could be a reflection of increased phosphate concentration, or decreased availability of labile carbon substrate. As the authors point out, this is peculiar and raises a lot of interesting questions. What are your thoughts based on your data? This is an important issue generally- how microbes deal with nutrient scarcity. It is probably not a bimodal response (one has enough or one doesn't). You have some important insights into this.

Likewise, *Trichodesmium* (if I remember correctly) is a potential source for phosphonates (paper in Nature by Dyhrman and Benitez-Nelson), but here you argue *Trichodesmium* is a potential sink for phosphonates. One of the striking bits of information in the recent Acker et al. paper on phosphonate production was that very few microbes both make and utilize phosphonates. If *Trichodesmium* is one of these, and apparently an important consumer, what is the likely advantage? How do you think this works?

I understand the length constraints that go along with the paper format, but I felt a lot of

potentially new ideas/contributions were missed in covering material that has already been covered by other papers.

More specific comments:

1) Although the distinction between potential net rates of methane production and rates of methane production was nicely made in the supplemental text, I think this distinction should be made for the reader in the paper proper. It does not detract from what the authors have done, but it is an important distinction, and it does (for example) impact your calculation of recycled P that can support production. I recommend that the abstract be changed to read:

“Here, we show high potential net rates of MPn-driven methane formation...”

And that the supplemental text:

“Our measured rates should be treated as potential net rates, as the methylphosphonate (MPn) additions ($1 \mu\text{mol L}^{-1}$) were in excess of the in-situ concentrations. In addition, potential methane consumption by aerobic oxidation was not measured here but previous measurements from the open ocean suggest these rates might be in the range of tens of picomolar^{76,77} and thus unlikely to strongly affect the reported rates. As the rates were mostly linear over 24 hours of incubation and often without a lag phase (Supplementary Figure S2), the microbial community appears to be well adapted to the utilisation of MPn or other phosphonates.”

be moved from the supplemental to the Results and Discussion, page 3, end of paragraph 1 (“into an area of high formation above the DCM and an area of low formation at the DCM and below it.”)

2) The Introduction- could be stronger. The purpose of the paper is framed as:

“Despite the important role of MPn utilisation in the marine carbon and phosphorus cycles, some of the most fundamental aspects of MPn-driven aerobic methane formation – such as depth distribution, magnitude and controlling physico-chemical factors – remain underexplored.”

This is very general. You might be more explicit about what you set out to explore. What are your hypotheses? What important questions were not addressed by previous papers? While indeed some of the parameters you mention are underexplored, we do have some information on them already.

3) Page 5 “In contrast to Pi, additions of nitrate were expected to increase rates of MPn-dependent methane formation, such as has been observed in the subtropical North Pacific^{10,32}.”

I forget exactly what Karl et al. did in the two citations, but I would expect N to stimulate methane production from MPn only when labile carbon substrate is in excess. The rationale for the N addition experiments should be described a bit better. What are the assumptions and what do the results show exactly?

I found it curious that the only significant enhancement in methane production rate for the MPn+N treatment was in the DCM. This is where Granzow et al. found the highest rates of C-P lyase activity, which presumably resulted from stronger P limitation. I don't recall if Granzow did a +N experiment, but as I wrote above, what is this all telling us about the “magnitude and controlling physico-chemical factors” of MPn?

4) Page 7 “However, out of the 96 incubations we were only able to detect one substantial rate (0.58 nmol CH₄ L⁻¹ d⁻¹), in the DCM of station S12 over the course of the 48 hours of incubation”

I think this is one of the more novel/interesting point of the paper and it merits a bit more discussion. Papers have shown that production of methane from MPn likely happens in the ocean, but since the original postulate on natural aerobic methane production from MPn a large number of other potential sources have been described, among which direct production through some unknown mechanism has gotten the most press. Here you show that this may actually happen outside of laboratory culture, but that, at least in this low P region of the ocean, it is probably not that common.

5) Page 8 (role of MPn..) I would change this to read “Therefore, our measured rates of potential net methane formation can be used to determine the potential (or maximum, or upper bound of..) amount of Pi liberated.....” or something like that. The range (0-100%) is quite large, and the mean (11%) is higher than the two other existing estimates based on methane flux and phosphate reduction. It might be interesting to discuss this a bit.

6) Supplemental text: Origin of methylphosphonate in surface waters.... I agree with all you have written however, my recollection is that (somewhat surprisingly) Acker et al. found very few sequences of archeal phosphonate producers in their metagenome analyses. I know number of reads does not equal activity, but....

I found the figures and tables to be easy to understand and helpful, as were the supplemental materials. I hope that the authors deposit their nutrient and related data (C/N/P ratios for POM for example) in a marine database.

Reviewer #3 (Remarks to the Author):

Overview:

This manuscript presents a well designed study into the production of methane from methyl-phosphonate in oligotrophic Atlantic seawater. Through a combination of isotope enrichment experiments and transcriptomics, the complex biogeochemistry involving methylphosphonate uptake is examined and the likely organisms responsible are identified

and the rates assessed. This work is well designed, performed and written. The authors should consider however the role of photochemistry on their data, given the comparison between methylphosphonate and DMS chemistry for methane production in the ocean, as detailed below.

General Comments:

Potential role of photochemistry/ROS in the degradation of methylphosphonate: One area that has apparently been overlooked in this work is the possibility of ^{13}C production from the ^{13}C labelled methylphosphonate due to photochemistry. It is known that such phosphonates can be photodegraded by UV (Yu et al., 2018; Zhang and Ji, 2019), most notably in the presence of iron (Lesueur et al., 2005). From laboratory studies it appears that the major photochemical pathway for the degradation of methylphosphonate is via indirect reactions with photochemically produced reactive oxygen species (ROS), most likely OH radicals (Xia et al., 2019). ROS species have also been suggested as a key driver of methane formation (Ernst et al., 2022) from biota. Work on Dimethyl Sulfide (DMS) has also shown that reaction with ROS is also an indirect photochemical pathway for methane formation in the ocean (Florez-Leiva et al., 2013), and that this photodegradation pathway is significant for DMS loss in the Ocean (Bouillon and Miller, 2004; Bouillon and Miller, 2005; Toole et al., 2003; Toole et al., 2004; Toole and Siegel, 2004; Toole et al., 2006). The photolysis loss rate is also increased in the presence of nitrate (Bouillon and Miller, 2005) due to the increased rate of OH formation.

In the context of the present work, it would be helpful then if this potential photochemical pathway was discussed and evaluated. For instance, inclusion of light/dark times for the methane formation plots (Figure S4) would help in this evaluation. Of interest here also is that it has been suggested that different pathways of methylphosphonate degradation (i.e. bacterial vs photodegradation) may be distinguished using the O-isotopes of the phosphate released (Yu et al., 2018).

Oxidation rate of ^{13}C during experiments: A recent paper (Mao et al., 2022) also studied the formation rate of methane from methylphosphonate as part of a wider study comparing potential CH_4 sources in marine shelf waters. In that work they found that methane oxidation rates were also quite significant, thus for the present work it would also be useful to include some comments on the potential losses due to oxidation of the ^{13}C produced and whether nitrate addition could have boosted both the production (e.g. hydroxyl radical

production and methylphosphonate degradation) and loss rates (e.g. photoreduction of nitrate to nitrite plus bacterial methane oxidation (Ettwig et al., 2010)).

Specific Comments:

Line 15: See the comment above regarding the potential for an indirect photochemical production pathway.

Line 25: methane (CH₄)

Line 66: The Deep Chlorophyll Maximum (DCM) is more typically described/modelled as a gaussian (Bouman et al., 2020; Lewis et al., 1983; Silulwane et al., 2001), than a parabolic (inverse).

Line 67: This is the sum of inorganic nitrogen species?

Line 69: It would be helpful here to include the detection limit that was found in this work (i.e. the value from line 493).

Line 110: See the general comment above regarding the potential for an indirect photochemical production pathway, as this would not be suppressed by the availability of inorganic phosphate.

Line 192: What is meant here by DIC as a precursor? Precursor for CH₄ formation? If so, is the rate then calculated from the formation of ¹³CH₄ from the ¹³C DIC addition experiments in the absence of other ¹³C sources?

Line 209: This is a very important aspect with regard to modelling of the P cycle in the ocean.

Line 219: Suggest authors use the term 'relative abundance' rather than abundance, as the data is normalized. If the total bacterial numbers are known also, this could be included here so the reader can get a picture of the overall number of phnJ containing organisms. Does this data also include Trichodesmium? See also Table S2 from Frischkorn et al. (2017) for the potential of the microbiome of Trichodesmium to contain phnJ.

Line 488: As the samples are frozen unfiltered, is there not the possibility of having cells break upon freezing, releasing nutrients?

Line 491: Please include the oxidation method used for the total phosphorus? Was this also a persulfate oxidation as for the POP? The Strickland and Parsons reference cited is a little out of date for such approaches now.

Line 500: Measured or calculated OP?

Line 507: Is the methodology here for both ¹²CH₄ and ¹³CH₄? It appears to be just ¹²CH₄

given the setup used.

Line 541: Thanks for including Figure S8, as it does indicate that there would be some UV reaching the near surface incubation bottles. See the general comment above regarding potential photochemical contributions.

References:

Bouillon, R.-C. and Miller, W.L., 2004. Determination of apparent quantum yield spectra of DMS photo-degradation in an in situ iron-induced Northeast Pacific Ocean bloom.

Geophysical Research Letters 31: L06310, doi:10.1029/2004GL019536.

Bouillon, R.C. and Miller, W.L., 2005. Photodegradation of dimethyl sulfide (DMS) in natural waters: Laboratory assessment of the nitrate-photolysis-induced DMS oxidation.

Environmental Science & Technology, 39(24): 9471-9477.

Bouman, H.A., Jackson, T., Sathyendranath, S. and Platt, T., 2020. Vertical structure in chlorophyll profiles: influence on primary production in the Arctic Ocean. *Philosophical Transactions of the Royal Society A: Mathematical, Physical and Engineering Sciences*, 378(2181): 20190351.

Ernst, L. et al., 2022. Methane formation driven by reactive oxygen species across all living organisms. *Nature*, 603(7901): 482-487.

Ettwig, K.F. et al., 2010. Nitrite-driven anaerobic methane oxidation by oxygenic bacteria. *Nature*, 464(7288): 543-548.

Florez-Leiva, L., Damm, E. and Farías, L., 2013. Methane production induced by dimethylsulfide in surface water of an upwelling ecosystem. *Progress in Oceanography*, 112–113: 38-48.

Frischkorn, K.R., Rouco, M., Van Mooy, B.A.S. and Dyhrman, S.T., 2017. Epibionts dominate metabolic functional potential of *Trichodesmium* colonies from the oligotrophic ocean. *The ISME Journal*, 11(9): 2090-2101.

Lesueur, C., Pfeffer, M. and Fuerhacker, M., 2005. Photodegradation of phosphonates in water. *Chemosphere*, 59(5): 685-691.

Lewis, M.R., Cullen, J.J. and Platt, T., 1983. Phytoplankton and thermal structure in the upper ocean: Consequences of nonuniformity in chlorophyll profile. *Journal of Geophysical Research: Oceans*, 88(C4): 2565-2570.

Mao, S.-H. et al., 2022. Aerobic oxidation of methane significantly reduces global diffusive methane emissions from shallow marine waters. *Nature Communications*, 13(1): 7309.

Silulwane, N.F., Richardson, A.J., Shillington, F.A. and Mitchell-Innes, B.A., 2001. Identification and classification of vertical chlorophyll patterns in the Benguela upwelling system and Angola-Benguela front using an artificial neural network. *South African Journal of Marine Science*, 23(1): 37-51.

Toole, D.A., Kieber, D.J., Kiene, R.P., Siegel, D.A. and Nelson, N.B., 2003. Photolysis and the dimethylsulfide (DMS) summer paradox in the Sargasso Sea. *Limnology and Oceanography*, 48: 1088-1100.

Toole, D.A. et al., 2004. High dimethylsulfide photolysis rates in nitrate-rich Antarctic waters. *Geophysical Research Letters*, 31, L11307, doi:10.1029/2004GL019863.

Toole, D.A. and Siegel, D.A., 2004. Light-driven cycling of dimethylsulfide (DMS) in the Sargasso Sea: Closing the loop. *Geophysical Research Letters*, 31: L09308, doi:10.1029/2004GL019581.

Toole, D.A., Slezak, D., Kiene, R.P., Kieber, D.J. and Siegel, D.A., 2006. Effects of solar radiation on dimethylsulfide cycling in the western Atlantic Ocean. *Deep Sea Research Part I: Oceanographic Research Papers*, 53(1): 136-153.

Xia, C. et al., 2019. Mechanism of methylphosphonic acid photo-degradation based on phosphate oxygen isotopes and density functional theory. *RSC Advances*, 9(54): 31325-31332.

Yu, C., Wang, F., Chang, S.J., Yao, J. and Blake, R.E., 2018. Phosphate oxygen isotope evidence for methylphosphonate sources of methane and dissolved inorganic phosphate. *Science of The Total Environment*, 644: 747-753.

Zhang, C. and Ji, H.B., 2019. Effects of environmental parameters on the ultraviolet degradation of methylphosphonate *Applied Ecology and Environmental Research*, 17(4): 9473-9482.

REVIEWER COMMENTS

Reviewer #1 (Remarks to the Author):

The study on methylphosphonate-driven methane formation by Arx et al. presented here is an interesting field study addressing which abiotic parameters and present bacterioplankton matter for methane formation. The results are especially needed to understand and predict better biogeochemical cycling in changing water bodies. In general, the manuscript is well-written and well-structured. However, I am unsure if it matches the broad readership of Nature Communications as it is quite detailed and field-specific (based on one field sampling campaign). The study might benefit from comparing two distinct field studies via, e.g. adding data from a field sampling elsewhere. The work will be mainly of interest to readers in the field of marine biogeochemistry and closely-related fields such as limnology and atmospheric sciences. I think the manuscript can benefit from including a few more explanatory sentences here and there. Concerning the methodology, I think apart from the molecular measurements (not my field of expertise), all used methods and analyses are sound and well-described in enough detail. I have one major issue, and a few minor issues outlined below. Otherwise, I enjoyed reading this manuscript, and hope to see it published soon.

We thank the reviewer for their comments. We acknowledge that we do not have a comparable dataset from a different field campaign that we can add to this manuscript, largely due to the complexity and time span of this type of seagoing field studies. Therefore, we have now used published field studies (e.g. from the North Pacific) for a more comprehensive comparison and explanation of our results, as suggested. We clarify that our research motivation was to understand the globally important process of microbial utilisation of organic phosphorus compounds and the associated aerobic methane production. We believe this knowledge is key for understanding the functioning of microbial communities under nutrient limitation and for accurate predictions of marine carbon and phosphorus cycling under future climate scenarios.

Major comment:

L. 69-71: A particulate organic N:P ratio of 20 indicates merely P-limitation, and absolutely not a strong P-limitation. See e.g. Ptacnik et al. (2010). I recommend revising this part and the discussion accordingly.

We believe this might be a misunderstanding - the respective sentence states that the inorganic nitrogen to Pi ratio increased below the DCM to excesses of 20 suggesting, based on Redfield ratio, that the water column was strongly Pi-limited. Above the DCM the inorganic nutrients were consistently at or below our detection limits and no ratios could be calculated. The particulate organic N:P ratio for all stations and depths was typically well above the Redfield ratio of 16:1 (data shown in Supplementary Figure S3). Our data are in line with historical observations from this area that identify the western Tropical North Atlantic as one of the most notoriously phosphate-limited oceanic regions (Galbraith and Martiny 2015; Moore et al. 2013; Sosa et al. 2019; Wu et al. 2000)

Nonetheless, we have now revised the manuscript to avoid references to strong P-limitation, as suggested.

Minor comments:

L. 17: Replace “such as” with e.g. including or most dominantly.

We have made this change.

L: 69-71: Reference for the Redfield ratio is needed, i.e., Redfield (1958), to aid the broad readership of Nature Communications.

We have added this reference.

L. 74: Please consider including the explanation here why the C-fixation rates declined 100-fold below the DCM.

We have included an explanation (disappearance of photosynthetically-active radiation and thus lower abundance of primary producers *in-situ* (Behrenfeld & Falkowski 1997; Cullen 2015) to the revised manuscript.

L. 127-128: Shorting header to increase readability.

We have shortened the header, which now reads:

“C-P lyase enzyme mainly encoded by *Trichodesmium* and *Alphaproteobacteria*”

L. 241-245: Please explain why a 10-m surface depth was selected, as often 1-m surface depth is used.

The explanation (technical restrictions, such as the height of rosette sampler + reproducibility due to e.g. wave action-induced turbulence) has been added to the manuscript together with the clarification that at each station, the 10-m surface depth was well within the mixed layer (Supplementary Figure S1).

Caption Fig. 1c: Please state that you determined the molar N:P ratio here.

We have changed this in both the Figure caption as well as the Figure itself.

L. 606-610: Please consider including a reference for RStudio software here for consistency.

We have added a reference.

Reviewer #2 (Remarks to the Author):

This study investigates the formation of methane in the eastern tropical/subtropical Atlantic Ocean, a region that is strongly phosphate limited and by inference a site where methane production from methylphosphonate is to be expected. The study makes several original contributions that are noteworthy: 1) the study provides the first potential net rate estimates of methane formation through the methylphosphonate (MP_{hn}) pathway for natural communities of marine microbes. 2) the study demonstrates that MP_{hn} is metabolised to methane even under relatively high (> 100 nM) phosphate concentrations supporting genomic and C-P lyase rate measurements that methane formation occurs in deeper, phosphate rich waters below the euphotic zone, 3) the study highlights a possible connection between nitrogen fixing Trichodesmium (or at least its microbiome) and methane formation, and 4) the study finds little evidence for direct production of methane from primary producers, suggesting at least in this limited study and at least in P-limited regions, direct production might not be that important a pathway. I found the manuscript to contribute original results to the topic of aerobic methane production in the ocean, which will influence and be influenced by climate change. The topic is a relatively new line of research in the field, and I therefore think the paper will be of broad interest to the community. The work was carefully done and the results and conclusions were sound.

We thank the reviewer for their comments.

Nevertheless, I felt the paper could be substantially improved with moderate revision by the authors. I think the paper would be more interesting and stimulating if the authors dug a bit deeper into the

questions addressed and by placing the data here within the data and conclusions reached by others (aside from a mere citation). Much of the discussion in the paper follows issues outlined in previous papers. The new measurements made by the authors offers a chance to more fully examine or discuss some of these questions, and I felt the authors missed an opportunity to do so.

We have now taken the opportunity to revise the discussion of our data both within the context of the available literature and the original research motivation (see below). We acknowledge that some relevant discussion points were in the Supplementary Discussion and we have now followed the advice of this and other reviewers to move and elaborate on them in the main text (e.g. the methane oxidation discussion and potential rates). Furthermore, we have added new discussion points as requested, such as the potential role of labile carbon, the rationale and effect of the nitrate additions as well as the methylphosphonate usage in the presence of phosphate.

For example, Holmes et al. GBC 2000 made estimates of methane flux to the atmosphere in the Sargasso Sea. Assuming (as Holmes et al does) that all of the methane produced in the euphotic zone is ventilated, at steady state how do your methane production values compare with Holmes et al.'s efflux data? There are several possible outcomes and interpretations if the two sets of measurements align or not.

If we integrate our potential net rates over the mixed layer, and omit lateral transport as well as downward flux, we get a total mean flux of $34.7 \pm 30.4 \mu\text{mol m}^{-2} \text{d}^{-1}$. For comparison, the concentration-based sea-air fluxes reported for the Sargasso Sea are around $4.4 \mu\text{mol m}^{-2} \text{d}^{-1}$ (Holmes et al. 2000) and for the ALOHA station around $1.4 - 1.7 \mu\text{mol m}^{-2} \text{d}^{-1}$ (Holmes et al. 2000). The order of magnitude difference between these fluxes can be explained by the fact that our potential rates very likely exceed the *in-situ* rates (due to excess substrate addition). We also calculated fluxes using our recorded wind speeds with the previously reported concentration measurements from this region (2.35 nmol L^{-1} at 5 m; Scranton and Brewer 1977) using the flux equation of (Holmes et al. 2000), which results in a mean flux of $4.8 \pm 1.9 \mu\text{mol m}^{-2} \text{d}^{-1}$ ($2.1 - 7.1 \mu\text{mol m}^{-2} \text{d}^{-1}$) for the duration of our sampling campaign.

We acknowledge that comparing rates to flux data is of interest as it can constrain how much this process contributes to atmospheric methane emissions. However, the scope of our study was to elucidate the mechanisms and rates of methane production, and the experimental design of our experiments was not well suited to quantify methane fluxes, which is why we refrained from doing so.

The authors point out that phosphonate degradation is regulated by the concentration of phosphate although the details are not known. Given your new data, there appears to be more you can say about this. As outlined on page 4, other studies have shown that likely rates of C-P lyase activity are high in and below the DCM, and genomic studies suggest the presence of C-P lyase (and other phosphonate utilisation pathway) genes and C-P lyase activity into the deep ocean. The repression in rate found here could be a reflection of increased phosphate concentration, or decreased availability of labile carbon substrate. As the authors point out, this is peculiar and raises a lot of interesting questions. What are your thoughts based on your data? This is an important issue generally- how microbes deal with nutrient scarcity. It is probably not a bimodal response (one has enough or one doesn't). You have some important insights into this.

We agree with the reviewer that identifying the mechanisms of phosphonate utilisation and regulation is essential to understand microbial response to nutrient scarcity. The reviewer's suggestion to consider availability of labile carbon is very interesting and we now discuss this point in the manuscript. We also prepared a new Supplementary Figure that depicts the relationship between methane production and selected parameters (Supplementary Figure S5).

Figure S5. **Methane formation from the methylphosphonate-addition experiment as a function of organic phosphorus (OP) concentrations (a), carbon fixation rates (b), phosphate (Pi) concentrations and particulate organic carbon concentrations (d).** The depths from all twelve stations were used: a 10-metre surface depth, intermediate depth, deep chlorophyll maximum (DCM) and below the DCM, except for S2 where nutrient data was unavailable. The mean rate of the triplicate carbon fixation incubations is presented in the graph, whereas the duplicates of the MPn incubation are shown separately.

1. Our data do not suggest that methane production coupled to phosphonate utilisation is regulated in a bimodal manner, i.e. is switched on/off by a certain threshold Pi concentration. While we consistently saw highest rates below Pi concentrations of 25 nmol L⁻¹ (Figure S5), we also observe that once Pi becomes detectable (which is a moving target, in our case 10 nmol L⁻¹), ‘low’ methane production (up to ca. 1 nmol L⁻¹ d⁻¹) is consistently detected even in the presence of high Pi concentrations (up to ca. 380 nmol L⁻¹ phosphate). This could be explained by a transient expression of the C-P lyase pathway even in the presence of Pi, due to:
 - a. The enzyme being expressed in order to aid the degradation of phosphonates, which might have cytotoxic effects, rather than for the purpose of obtaining P.
 - b. Presence of specialist MPn/phosphonate utilisers. A group of organisms could be specialised in degrading MPn or other phosphonates to avoid competition with phosphate utilising organisms and possibly gaining additional benefits such as using MPn as a carbon source.
 - c. Persistent MPn utilisation. *Trichodesmium* can metabolise MPn and Pi at almost equal metabolic efficiency (Beverdors et al. 2010) and this feature might be present in other organisms as well.

Hence, we hesitate to propose a “new” Pi threshold for this process as it may be defined by a combination of physicochemical (e.g. OP/Pi ratios, N/C/Fe availability) and biological (dominant MPn utilising taxa) parameters, and would thus not be universally applicable across different oceanic regions. Our data nonetheless confirm the prevailing view that this process tends to get “repressed” at deeper depths.

2. Methane production did not increase proportionally to the rates of primary production, suggesting that labile DOC (produced by PP) is not controlling the magnitude of this process, at

least not in surface waters (see also answer to Specific comment 3). The methane production rates also did not increase proportionally to the availability of OP, suggesting that the process does not simply respond to substrate availability. However, OP does not equal phosphonates (or MPn specifically) and we would need more data on MPn or phosphonate *in-situ* concentrations in order to constrain this regulating factor more closely.

Taken together, it appears that in excess of labile carbon, non-detectable Pi concentrations and presence of OP higher than ca. 94 nmol L⁻¹ - such as in the surface depths - the magnitude of MPn-driven methane production rate is mainly regulated by the phosphate (un)availability. This is supported by our MPn-driven rates but also our Pi addition experiments. At and below the DCM labile carbon might be less available, even limiting, the POC is lower and the OP is generally lower than at the surface. Under these conditions, Pi does not appear to exert main control over this process. This is supported by our experiments where addition of Pi hardly affected the magnitude of the rates at these depths. As these are - to our knowledge - the first rates measured from below the DCM we look forward to test these hypotheses in our future work.

Likewise, *Trichodesmium* (if I remember correctly) is a potential source for phosphonates (paper in Nature by Dyhrman and Benitez-Nelson), but here you argue *Trichodesmium* is a potential sink for phosphonates. One of the striking bits of information in the recent Acker et al. paper on phosphonate production was that very few microbes both make and utilize phosphonates. If *Trichodesmium* is one of these, and apparently an important consumer, what is the likely advantage? How do you think this works?

We agree that this is important to clarify, especially given the key role of *Trichodesmium* in the western Tropical North Atlantic as a primary producer, nitrogen fixer and MPn utiliser. The ability of the *Trichodesmium* genus to use phosphonates and specifically MPn is well established in the literature (Beverdors et al. 2010; Dyhrman et al. 2006), and we focus on this role because our study aimed to identify organisms with the capacity to use methylphosphonates (and produce methane). But the reviewer is right and *Trichodesmium* can also synthesise phosphonates (Dyhrman et al. 2009; Frischkorn et al. 2017, 2018) although it does not seem to be able to produce MPn specifically and neither can its microbiome (Frischkorn et al. 2018). We did not recover *mpnS* gene sequences from *Trichodesmium* from our samples (data not shown) and therefore cannot comment on whether *Trichodesmium* was contributing to MPn production in our study area.

The capacity to both make and utilise phosphonates indeed seems to be rare, but it is present in abundant keystone marine lineages, including *Trichodesmium* and *SAR11* (Born et al. 2017; Carini et al. 2014) and more such taxa may be uncovered in the future. In *Trichodesmium* specifically, there may be several explanations for the coexistence of these pathways (e.g. spatial and temporal separation, 'luxury' metabolism and competition), which we now elaborate on in the Supplementary Discussion.

I understand the length constraints that go along with the paper format, but I felt a lot of potentially new ideas/contributions were missed in covering material that has already been covered by other papers.

More specific comments:

1) Although the distinction between potential net rates of methane production and rates of methane production was nicely made in the supplemental text, I think this distinction should be made for the reader in the paper proper. It does not detract from what the authors have done, but it is an important distinction, and it does (for example) impact your calculation of recycled P that can support production. I recommend that the abstract be changed to read:

"Here, we show high potential net rates of MPn-driven methane formation...."

And that the supplemental text:

"Our measured rates should be treated as potential net rates, as the methylphosphonate (MPn) additions ($1 \mu\text{mol L}^{-1}$) were in excess of the in-situ concentrations. In addition, potential methane consumption by aerobic oxidation was not measured here but previous measurements from the open ocean suggest these rates might be in the range of tens of picomolar^{76,77} and thus unlikely to strongly affect the reported rates. As the rates were mostly linear over 24 hours of incubation and often without a lag phase (Supplementary Figure S2), the microbial community appears to be well adapted to the utilisation of MPn or other phosphonates."

be moved from the supplemental to the Results and Discussion, page 3, end of paragraph 1 ("into an area of high formation above the DCM and an area of low formation at the DCM and below it."

We have moved these paragraphs accordingly.

2) The Introduction- could be stronger. The purpose of the paper is framed as:

“Despite the important role of MPn utilisation in the marine carbon and phosphorus cycles, some of the most fundamental aspects of MPn-driven aerobic methane formation – such as depth distribution, magnitude and controlling physico-chemical factors – remain underexplored.

This is very general. You might be more explicit about what you set out to explore. What are your hypotheses? What important questions were not addressed by previous papers? While indeed some of the parameters you mention are underexplored, we do have some information on them already.

We thank the reviewer for this suggestion. We have now elaborated in more detail the specific questions we wanted to address with our study. The Introduction now reads:

“Despite the important role of MPn utilisation in the marine carbon and phosphorus cycles, the process has so far been experimentally confirmed only in a handful of oceanic regions, mainly around station ALOHA in the eastern Tropical North Pacific (Del Valle & Karl 2014; Karl et al. 2008; Repeta et al. 2016) and in the Sargasso Sea (Sosa et al. 2020). To date, little is known about the depth distribution of the process in the water column and the physico-chemical factors that control it.”

3) Page 5 “In contrast to Pi, additions of nitrate were expected to increase rates of MPn-dependent methane formation, such as has been observed in the subtropical North Pacific^{10,32}.”

I forget exactly what Karl et al. did in the two citations, but I would expect N to stimulate methane production from MPn only when labile carbon substrate is in excess. The rationale for the N addition experiments should be described a bit better. What are the assumptions and what do the results show exactly?

We thank the reviewer for their comment and elaborate more on the aims and results of the nitrate addition experiment. Briefly, nitrate was expected to stimulate MPn utilisation both directly and indirectly - directly, as MPn consumers in oligotrophic environments might be N-limited, such as proposed before for the ETNP (Del Valle & Karl 2014); indirectly, as additions of nitrate are expected to exacerbate P stress and thus promote utilisation of MPn. In prior experiments, nitrate additions enhanced the overall rates of MPn utilisation, however, this was done in waters that were not completely Pi-depleted (ETNP; Karl et al. 2008). Our rate stimulation was slightly higher than those observed by (Del Valle & Karl 2014) and we speculate that the effect might become even more pronounced under longer incubation times (e.g. 60 hours of incubation observed by Karl et al. 2008).

Additionally, both the heterotrophic as well as the autotrophic productivity in the Atlantic seems to be N and P limited (Mills et al. 2008; Moore et al. 2008); therefore, the addition of nitrate and MPn in the Redfield ratio was expected to enhance microbial growth. We did not expect labile DOC in the surface waters to be limiting as around few to tens of $\mu\text{mol C/kg}$ are produced and renewed by autotrophs daily (Carlson & Hansell 2015). On second thought, it is possible that availability of labile DOM affects the rate of MPn utilisation deeper in the water column, where dominant MPn utilisers are mixo- or heterotrophic. We now discuss this possibility in the text.

I found it curious that the only significant enhancement in methane production rate for the MPn+N treatment was in the DCM. This is where Granzow et al. found the highest rates of C-P lyase activity, which presumably resulted from stronger P limitation. I don't recall if Granzow did a +N experiment, but as I wrote above, what is this all telling us about the "magnitude and controlling physico-chemical factors" of MPn?

We now elaborate more on the potential controlling parameters in our manuscript. Briefly, we speculate that (i) high rates of MPn utilisation are a response to an acute lack of P_i and can thus be stimulated by nitrogen (that tends to exacerbate P stress) and significantly repressed by inorganic phosphate; while (ii) low 'background' rates of MPn utilisation seem to persist in phosphate-replete environments, possibly due to serving other purposes than solely P acquisition (see also answer to an earlier comment).

Unfortunately, (Granzow et al. 2021) do not report N-addition experiments in their manuscript. However, our results support the observation of Granzow et al. and others that MPn-driven methane can be produced below the surface mixed layer; even though in contrast to Granzow et al. we consistently detected highest rates in the surface depths and not in the DCM.

We should point out that the stimulating effect of nitrate in the DCM was the only one that was statistically significant, but the medians were elevated in every depth. The lack of a statistically significant effect in the upper two incubation depths may be due to the overall more heterogeneous nature of the rates observed. This is likely due to the contribution of the larger *Trichodesmium* that was most abundant in the upper two incubation depths as well. Therefore, the DCM itself may not be the sole hotspot for the rate stimulation by nitrate additions. We have now added an explanatory sentence to the manuscript regarding this.

4) Page 7 “However, out of the 96 incubations we were only able to detect one substantial rate (0.58 nmol CH₄ L⁻¹ d⁻¹), in the DCM of station S12 over the course of the 48 hours of incubation”

I think this is one of the more novel/interesting point of the paper and it merits a bit more discussion. Papers have shown that production of methane from MPn likely happens in the ocean, but since the original postulate on natural aerobic methane production from MPn a large number of other potential sources have been described, among which direct production through some unknown mechanism has gotten the most press. Here you show that this may actually happen outside of laboratory culture, but that, at least in this low P region of the ocean, it is probably not that common.

Indeed, one of our interests for our study was to address the possibility that photosynthesis may contribute to aerobic methane production. However, based on our data, aerobic methane production in this region is dominated by MPn-driven methane, while any DIC-driven methane formation is either hugely sporadic or proceeds at much lower rates than that of MPn.

Unfortunately, our capacity to detect DIC-driven methane *in-situ* suffers from the low maximum labelling percentage that we can use (in order not to alter the pH of the incubations), thus increasing our limit of detection. At the same time, if we use longer incubation times, we will not be able to say with confidence whether methane was produced directly from DIC, or from any of the many carbon intermediates that are formed during autotrophic cell metabolism. Additionally, to our knowledge, the pathway(s) for aerobic DIC-based methane production are not conclusively resolved on a biochemical and genetic level, which precludes us from relying on molecular analyses.

We have now elaborated more on the possible role of DIC-driven methane.

5) Page 8 (role of MPn..) I would change this to read “Therefore, our measured rates of potential net methane formation can be used to determine the potential (or maximum, or upper bound of..) amount of Pi liberated.....” or something like that. The range (0-100%) is quite large, and the mean (11%) is higher than the two other existing estimates based on methane flux and phosphate reduction. It might be interesting to discuss this a bit.

We have now reformulated this as suggested.

Regarding the contribution estimate - we want to clarify that this is an estimate of the (maximal) contribution of P released from phosphonate degradation for primary productivity. We are not aware of

other quantitative estimates regarding C-P lyase activity and primary productivity but interestingly, our median (11%) is at the lower end of the estimate for DOP driven primary productivity through the degradation with alkaline phosphatases (12 - 30% in the North Atlantic subtropical gyre; Mather et al. 2008). We have now added this to the main text.

We have now also included a discussion of phosphorus redox cycling (Van Mooy et al. 2015) as well as the link between phosphonate turnover and methane production (Repeta et al. 2016) in the respective paragraphs.

6) Supplemental text: Origin of methylphosphonate in surface waters.... I agree with all you have written however, my recollection is that (somewhat surprisingly) Acker et al. found very few sequences of archeal phosphonate producers in their metagenome analyses. I know number of reads does not equal activity, but....

We too think this is an important issue to raise. MPn synthesis still is vastly underinvestigated and underrepresented in the literature compared to the C-P lyase pathway and future research should aim to address this discrepancy.

More specifically to the findings of Acker et al. (2022): To our understanding they looked at general phosphonate synthesis (by focusing on the *pepM* gene) and not on MPn synthesis specifically (by looking at the *mpnS*), although it was included in their analyses. Lockwood et al. (2022) looked at the *mpnS* more specifically and found that it was present in 7.7% of the community of the mesopelagic ocean and in less than 1.5% in the surface and DCM communities, although their transcripts remained below 0.25 per 100.000 reads. Their Figure 2 also shows that *Thaumarchaeota* can make up a significant fraction of the *mpnS*-containing community, especially compared to other genes investigated. We have included this discussion in the supplementary text of the revised manuscript.

I found the figures and tables to be easy to understand and helpful, as were the supplemental materials. I hope that the authors deposit their nutrient and related data (C/N/P ratios for POM for example) in a marine database.

The *phnJ* sequences have been deposited to NCBI under accession numbers OQ435733-68, as stated in the manuscript and will be available upon publication. Source data for the figures will be available on

Figshare, similarly, station, nutrient and rate data are planned to be deposited to PANGAEA (PANGAEA® - Data Publisher for Earth & Environmental Science doi:10.1594/PANGAEA).

Reviewer #3 (Remarks to the Author):

Overview:

This manuscript presents a well designed study into the production of methane from methylphosphonate in oligotrophic Atlantic seawater. Through a combination of isotope enrichment experiments and transcriptomics, the complex biogeochemistry involving methylphosphonate uptake is examined and the likely organisms responsible are identified and the rates assessed. This work is well designed, performed and written. The authors should consider however the role of photochemistry on their data, given the comparison between methylphosphonate and DMS chemistry for methane production in the ocean, as detailed below.

General Comments:

Potential role of photochemistry/ROS in the degradation of methylphosphonate: One area that has apparently been overlooked in this work is the possibility of ^{13}C production from the ^{13}C labelled methylphosphonate due to photochemistry. It is known that such phosphonates can be photodegraded by UV (Yu et al., 2018; Zhang and Ji, 2019), most notably in the presence of iron (Lesueur et al., 2005). From laboratory studies it appears that the major photochemical pathway for the degradation of methylphosphonate is via indirect reactions with photochemically produced reactive oxygen species (ROS), most likely OH radicals (Xia et al., 2019). ROS species have also been suggested as a key driver of methane formation (Ernst et al., 2022) from biota. Work on Dimethyl Sulfide (DMS) has also shown that reaction with ROS is also an indirect photochemical pathway for methane formation in the ocean (Florez-Leiva et al., 2013), and that this photodegradation pathway is significant for DMS loss in the Ocean (Bouillon and Miller, 2004; Bouillon and Miller, 2005; Toole et al., 2003; Toole et al., 2004; Toole and Siegel, 2004; Toole et al., 2006). The photolysis loss rate is also increased in the presence of nitrate (Bouillon and Miller, 2005) due to the increased rate of OH formation.

In the context of the present work, it would be helpful then if this potential photochemical pathway was discussed and evaluated. For instance, inclusion of light/dark times for the methane formation

plots (Figure S4) would help in this evaluation. Of interest here also is that it has been suggested that different pathways of methylphosphonate degradation (i.e. bacterial vs photodegradation may be distinguished using the O-isotopes of the phosphate released (Yu et al., 2018).

We thank the reviewer for their comment. We acknowledge that we had not discussed the potential role of photochemistry and we have now included this point in both the introduction and discussion. While we did not specifically test for photochemical degradation of MPn, we argue that the contribution of any such process is likely marginal at best and will not affect our original conclusions. This is due to:

I) Our incubations (48 hours) covered two day and night cycles (with up to 12 h 15 minutes of darkness), whereas methane production proceeded linearly over time, without any clear relation to light availability. As suggested by the reviewer, we have now indicated this in the Supplementary Figure S4 which depicts the time course of methane formation.

II) Abiotic methane formation would be expected to produce similar amounts of methane in incubations exposed to the same light conditions and containing the same amount of ^{13}C -MPn tracer. This was not the case. Methane production in the surface and intermediate samples (which were incubated in the same incubator) varied by one to two orders of magnitude, as did even duplicate samples from the same depths. Additionally, water samples, to which phosphate was added had significantly lower methane production rates than the MPn incubations only, even though they were set up from the same water, with the same amount of MPn and incubated under the same light conditions.

III) The conditions under which photochemical MPn degradation was shown to occur are not comparable to the conditions in our incubations or *in-situ*. Our rates are calculated from 24 hour incubations, compared to the 2100 hour incubation times of Yu et al. (2018) and 58 hours of Zhang & Ji (2019). Furthermore, most efficient degradation was observed at both highly alkaline and acidic conditions (Zhang & Ji 2019). MPn decomposition was only observed at UV intensity of 1200W, with little to no effect at 400W, and a UV lamp was used instead of natural light (Zhang & Ji 2019).

In-situ, the UV penetration depth Z10% for both UV-A and UV-B likely does not exceed 35 m (Tedetti & Sempéré 2006 and references therein) and thus would not affect 3 out of the 4 incubation depths.

Taken together, we are convinced that the ^{13}C -methane that we measure in our incubations is a result of biological turnover of MPn and not photochemical degradation. Nonetheless, it would surely be

worthwhile to test for photochemical MPn degradation under a range of environmentally-relevant settings (pH, irradiation intensity, temperature, time). We have now included these considerations into the supplementary text of the revised manuscript

Oxidation rate of $^{13}\text{CH}_4$ during experiments: A recent paper (Mao et al., 2022) also studied the formation rate of methane from methylphosphonate as part of a wider study comparing potential CH_4 sources in marine shelf waters. In that work they found that methane oxidation rates were also quite significant, thus for the present work it would also be useful to include some comments on the potential losses due to oxidation of the $^{13}\text{CH}_4$ produced and whether nitrate addition could have boosted both the production (e.g. hydroxyl radical production and methylphosphonate degradation) and loss rates (e.g. photoreduction of nitrate to nitrite plus bacterial methane oxidation (Ettwig et al., 2010)).

We agree that the potential mitigating effect of aerobic methane oxidation is important to discuss and we have now moved the respective paragraph from the supplementary text to the main body.

Unfortunately, open ocean methane oxidation rates tend to be so low that they can only be reliably detected with ^{14}C or tritium and we did not have the possibility to do radiotracer work on our cruise. This also contributed to our decision to focus our study on the mechanisms of methane production rather than on overall methane cycling and fluxes.

The point about nitrate potentially stimulating methane oxidation raised by the reviewer is interesting and we have now added a sentence about this possibility when discussing the effects of the nitrate addition. We also discuss the possibility of anaerobic methane oxidation occurring in anoxic microniches.

Specific Comments:

Line 15: See the comment above regarding the potential for an indirect photochemical production pathway.

We have amended the introduction accordingly.

Line 25: methane (CH_4)

This has been added.

Line 66: The Deep Chlorophyll Maximum (DCM) is more typically described/modelled as a gaussian (Bouman et al., 2020; Lewis et al., 1983; Silulwane et al., 2001), than a parabolic (inverse).

We have now changed this accordingly.

Line 67: This is the sum of inorganic nitrogen species?

Yes, that is correct. We have now revised this sentence to read:

‘Inorganic N (nitrogen oxides + ammonium) to P (Pi) ratios were not....’

Line 69: It would be helpful here to include the detection limit that was found in this work (i.e. the value from line 493).

This has now been added.

Line 110: See the general comment above regarding the potential for an indirect photochemical production pathway, as this would not be suppressed by the availability of inorganic phosphate.

As outlined above, we are convinced that the MPn degradation to methane is biological and the lack of effect of Pi addition to deeper depths is most likely due to differential regulation. Please also see answer to Reviewer 2.

Line 192: What is meant here by DIC as a precursor? Precursor for CH₄ formation? If so, is the rate then calculated from the formation of ¹³CH₄ from the ¹³C DIC addition experiments in the absence of other ¹³C sources?

Yes, that is correct. We have now changed the wording to read:

We also investigated the possibility that DIC can be converted to methane in the investigated study site, for which we set up parallel experiments in duplicate using only ¹³C-DIC.

Line 209: This is a very important aspect with regard to modelling of the P cycle in the ocean.

We agree with this comment and correspondingly we have now emphasised this potential aspect more strongly in the introduction as well, as we think the potential implications are of interest to a broad range of fields.

Line 219: Suggest authors use the term ‘relative abundance’ rather than abundance, as the data is normalized. If the total bacterial numbers are known also, this could be included here so the reader can get a picture of the overall number of *phnJ* containing organisms. Does this data also include *Trichodesmium*? See also Table S2 from Frischkorn et al. (2017) for the potential of the microbiome of *Trichodesmium* to contain *phnJ*.

Yes, the data in Table 1 also include *Trichodesmium*. We have retrieved *phnJ* genes belonging to *Trichodesmium* from the >10 um size fraction. The phylogenetic placement of the *Trichodesmium phnJ* sequence is shown in Figure 4 and the TPM values are listed in the Figure legend.

Additionally, we now also mention the possibility that some of the bacteria containing *phnJ* gene (e.g. the *Rhodospirillales* and *Gammaproteobacteria*) may constitute part of the *Trichodesmium* microbiome *in-situ* (as shown by Frischkorn et al. 2017), rather than be free-living. Unfortunately, we do not have the total bacterial numbers therefore we have now changed ‘abundance’ to ‘relative abundance’, as suggested.

Line 488: As the samples are frozen unfiltered, is there not the possibility of having cells break upon freezing, releasing nutrients?

That is indeed possible, and we now mention this possibility in the text. Unfortunately, sterile-filtering the samples has its own drawbacks, as pointed out e.g. by (Marvin et al. 1972).

Nevertheless, the inorganic nutrient concentrations (P_i , as well as NO_x) in the surface waters where the cell biomass is highest remained below detection limit; therefore, we argue that any such intracellular nutrient release would have been negligible.

Line 491: Please include the oxidation method used for the total phosphorus? Was this also a persulfate oxidation as for the POP? The Strickland and Parsons reference cited is a little out of date for such approaches now.

We have now included the protocol used for total phosphorus measurements.

Line 500: Measured or calculated

The OP was calculated. We have now clarified this.

Line 507: Is the methodology here for both $^{12}\text{C-CH}_4$ and $^{13}\text{C-CH}_4$? It appears to be just $^{12}\text{C-CH}_4$ given the setup used.

This section refers to the *in situ* $^{12}\text{C-CH}_4$ concentration measurements, which were done on untreated water samples and for which no data are shown because of unknown interference/contamination in the samples. These were taken independently from the incubation experiments, in which $^{13}\text{C-CH}_4$ production was measured. Both the header and the first line have now been amended for clarity to read:

Dissolved in-situ methane concentration measurements

Duplicate samples for methane concentrations...

Line 541: Thanks for including Figure S8, as it does indicate that there would be some UV reaching the near surface incubation bottles. See the general comment above regarding potential photochemical contributions.

We now discuss this point in the revised manuscript. See also response to earlier comment.

References

Acker M, Hogle SL, Berube PM, Hackl T, Coe A, et al. 2022. Phosphonate production by marine microbes: Exploring new sources and potential function. *Proc. Natl. Acad. Sci. U. S. A.* 119(11):

Behrenfeld MJ, Falkowski PG. 1997. Photosynthetic rates derived from satellite-based chlorophyll concentration. *Limnol. Oceanogr.* 42(1):1–20

Beverdorsdorf LJ, White AE, Björkman KM, Letelier RM, Karl DM. 2010. Phosphonate metabolism by *Trichodesmium* IMS101 and the production of greenhouse gases. *Limnol. Oceanogr.* 55(4):1768–78

Born DA, Ulrich EC, Ju KS, Peck SC, Van Der Donk WA, Drennan CL. 2017. Structural basis for methylphosphonate biosynthesis. *Science (80-.).* 358(6368):1336–39

Carini P, White AE, Campbell EO, Giovannoni SJ. 2014. Methane production by phosphate-starved SAR11 chemoheterotrophic marine bacteria. *Nat. Commun.* 5:1–7

Carlson CA, Hansell DA. 2015. DOM Sources, Sinks, Reactivity, and Budgets. In *Biogeochemistry of Marine Dissolved Organic Matter*, pp. 65–126. Elsevier

- Cullen JJ. 2015. Subsurface chlorophyll maximum layers: Enduring enigma or mystery solved? *Ann. Rev. Mar. Sci.* 7:207–39
- Del Valle DA, Karl DM. 2014. Aerobic production of methane from dissolved water-column methylphosphonate and sinking particles in the North Pacific Subtropical Gyre. *Aquat. Microb. Ecol.* 73(2):93–105
- Dyhrman ST, Benitez-Nelson CR, Orchard ED, Haley ST, Pellechia PJ. 2009. A microbial source of phosphonates in oligotrophic marine systems. *Nat. Geosci.* 2(10):696–99
- Dyhrman ST, Chappell PD, Haley ST, Moffett JW, Orchard ED, et al. 2006. Phosphonate utilization by the globally important marine diazotroph *Trichodesmium*. *Nature.* 439(7072):68–71
- Frischkorn KR, Krupke A, Guieu C, Louis J, Rouco M, et al. 2018. *Trichodesmium* physiological ecology and phosphate reduction in the western tropical South Pacific. *Biogeosciences.* 15(19):5761–78
- Frischkorn KR, Rouco M, Van Mooy BAS, Dyhrman ST. 2017. Epibionts dominate metabolic functional potential of *Trichodesmium* colonies from the oligotrophic ocean. *ISME J.* 11(9):2090–2101
- Galbraith ED, Martiny AC. 2015. A simple nutrient-dependence mechanism for predicting the stoichiometry of marine ecosystems. *Proc. Natl. Acad. Sci. U. S. A.* 112(27):8199–8204
- Granzow BN, Sosa OA, Gonnelli M, Santinelli C, Karl DM, Repeta DJ. 2021. A sensitive fluorescent assay for measuring carbon-phosphorus lyase activity in aquatic systems. *Limnol. Oceanogr. Methods.* 19(4):235–44
- Holmes ME, Sansone FJ, Rust TM, Popp BN. 2000. Methane production, consumption, and air-sea exchange in the open ocean: An Evaluation based on carbon isotopic ratios. *Global Biogeochem. Cycles.* 14(1):1–10
- Karl DM, Beversdorf L, Björkman KM, Church MJ, Martinez A, Delong EF. 2008. Aerobic production of methane in the sea. *Nat. Geosci.* 1(7):473–78
- Lockwood S, Greening C, Baltar F, Morales SE. 2022. Global and seasonal variation of marine phosphonate metabolism. *ISME J.* 16(9):2198–2212

- Marvin KT, Proctor RR, Neal RA. 1972. SOME EFFECTS OF FILTRATION ON THE DETERMINATION OF NUTRIENTS IN FRESH AND SALT WATER. *Limnol. Oceanogr.* 17(5):777–84
- Mather RL, Reynolds SE, Wolff GA, Williams RG, Torres-Valdes S, et al. 2008. Phosphorus cycling in the North and South Atlantic Ocean subtropical gyres. *Nat. Geosci.* 1(7):439–43
- Mills MM, Moore CM, Langlois R, Milne A, Achterberg E, et al. 2008. Nitrogen and phosphorus co-limitation of bacterial productivity and growth in the oligotrophic subtropical North Atlantic. *Limnol. Oceanogr.* 53(2):824–34
- Moore CM, Mills MM, Arrigo KR, Berman-Frank I, Bopp L, et al. 2013. Processes and patterns of oceanic nutrient limitation. *Nat. Geosci.* 6(9):701–10
- Moore CM, Mills MM, Langlois R, Milne A, Achterberg EP, et al. 2008. Relative influence of nitrogen and phosphorus availability on phytoplankton physiology and productivity in the oligotrophic subtropical North Atlantic Ocean. *Limnol. Oceanogr.* 53(1):291–305
- Repeta DJ, Ferrón S, Sosa OA, Johnson CG, Repeta LD, et al. 2016. Marine methane paradox explained by bacterial degradation of dissolved organic matter. *Nat. Geosci.* 9(12):884–87
- Scranton MI, Brewer PG. 1977. Occurrence of methane in the near-surface waters of the western subtropical North-Atlantic. *Deep. Res.* 24(2):127–38
- Sosa OA, Burrell TJ, Wilson ST, Foreman RK, Karl DM, Repeta DJ. 2020. Phosphonate cycling supports methane and ethylene supersaturation in the phosphate-depleted western North Atlantic Ocean. *Limnol. Oceanogr.* 65(10):2443–59
- Sosa OA, Repeta DJ, DeLong EF, Ashkezari MD, Karl DM. 2019. Phosphate-limited ocean regions select for bacterial populations enriched in the carbon–phosphorus lyase pathway for phosphonate degradation. *Environ. Microbiol.* 21(7):2402–14
- Tedetti M, Sempéré R. 2006. Penetration of Ultraviolet Radiation in the Marine Environment. A Review. *Photochem. Photobiol.* 82(2):389
- Van Mooy BAS, Krupke A, Dyhrman ST, Fredricks HF, Frischkorn KR, et al. 2015. Major role of planktonic phosphate reduction in the marine phosphorus redox cycle. *Science (80-.).* 348(6236):783–85

Wu J, Sunda W, Boyle EA, Karl DM. 2000. Phosphate depletion in the Western North Atlantic Ocean. *Science (80-.).* 289(5480):759–62

Yu C, Wang F, Chang SJ, Yao J, Blake RE. 2018. Phosphate oxygen isotope evidence for methylphosphonate sources of methane and dissolved inorganic phosphate. *Sci. Total Environ.* 644:747–53

Zhang C, Ji HB. 2019. Effects of environmental parameters on the ultraviolet degradation of methylphosphonate. *Appl. Ecol. Environ. Res.* 17(4):9473–82

REVIEWERS' COMMENTS

Reviewer #2 (Remarks to the Author):

The authors did a terrific job in addressing all my concerns with the paper. The responses and edits were thoughtful and I appreciate that the authors commitment to improving the paper. I congratulate them on a very nice study and such an interesting paper. I only have two small editorial comments:

Page 6, first paragraph. Change “Interestingly, in the below DCM depth...” to “Interestingly, in depths below the DCM....”

Page 9, second paragraph (Role of MPn...) change “As, methane and Pi...” to “As methane and Pi...” (remove comma)

Other than these, I am happy to see the paper accepted without further revision.

Reviewer #3 (Remarks to the Author):

The authors have made a good job of revising the manuscript and addressing the comments of myself and the other reviewers. I now find that the paper is acceptable for publication.

Reviewer #2 (Remarks to the Author):

The authors did a terrific job in addressing all my concerns with the paper. The responses and edits were thoughtful and I appreciate that the authors commitment to improving the paper. I congratulate them on a very nice study and such an interesting paper. I only have two small editorial comments:

We would like to thank the reviewer for the kind reply. We are also thankful for their earlier comments that helped improve our study.

Page 6, first paragraph. Change “Interestingly, in the below DCM depth...” to “Interestingly, in depths below the DCM....”

We have changed this as suggested.

Page 9, second paragraph (Role of MPn...) change “As, methane and Pi...” to “As methane and Pi...” (remove comma)

We have corrected this mistake.

Other than these, I am happy to see the paper accepted without further revision.

Reviewer #3 (Remarks to the Author):

The authors have made a good job of revising the manuscript and addressing the comments of myself and the other reviewers. I now find that the paper is acceptable for publication.

We would like to thank the reviewer for their kind comment and for their thoughtful earlier suggestions that helped improve our manuscript.